# Comprehensive single-cell sequencing reveals the stromal dynamics and tumor-specific characteristics in the microenvironment of nasopharyngeal carcinoma

Lanqi Gong[1,2], Dora Lai-Wan Kwong[1,2], Wei Dai [1,2], Pingan Wu[3], Shanshan Li[1], Qian Yan[1,2], Yu Zhang[2], Baifeng Zhang [2], Xiaona Fang[2], Li Liu [4,5,6], Min Luo[1], Beilei Liu [2], Larry Ka-Yue Chow[2], Qingyun Chen[7], Jinlin Huang[2], Victor Ho-Fun Lee[1,2], Ka-On Lam [1,2], Anthony Wing-Ip Lo [8], Zhiwei Chen [4,5,6], Yan Wang[9], Anne Wing-Mui Lee[1,2] & Xin-Yuan Guan [1,2,7 ✉]

The tumor microenvironment (TME) of nasopharyngeal carcinoma (NPC) harbors a heterogeneous and dynamic stromal population. A comprehensive understanding of this tumor-specific ecosystem is necessary to enhance cancer diagnosis, therapeutics, and prognosis. However, recent advances based on bulk RNA sequencing remain insufficient to construct an in-depth landscape of infiltrating stromal cells in NPC. Here we apply single-cell RNA sequencing to 66,627 cells from 14 patients, integrated with clonotype identification on T and B cells. We identify and characterize five major stromal clusters and 36 distinct subpopulations based on genetic profiling. By comparing with the infiltrating cells in the non-malignant microenvironment, we report highly representative features in the TME, including phenotypic abundance, genetic alternations, immune dynamics, clonal expansion, developmental trajectory, and molecular interactions that profoundly influence patient prognosis and therapeutic outcome. The key findings are further independently validated in two single-cell RNA sequencing cohorts and two bulk RNA-sequencing cohorts. In the present study, we reveal the correlation between NPC-specific characteristics and progression-free survival. Together, these data facilitate the understanding of the stromal landscape and immune dynamics in NPC patients and provides deeper insights into the development of prognostic biomarkers and therapeutic targets in the TME.

[1] Department of Clinical Oncology, The University of Hong Kong-Shenzhen Hospital, Shenzhen, China. [2] Department of Clinical Oncology, Li Ka Shing Faculty of Medicine, The University of Hong Kong, Hong Kong, China. [3] Department of Surgery, The University of Hong Kong-Shenzhen Hospital, Shenzhen, China. [4] Department of Microbiology, Li Ka Shing Faculty of Medicine, The University of Hong Kong, Hong Kong, China. [5] The AIDS Institute, Li Ka Shing Faculty of Medicine, The University of Hong Kong, Hong Kong, China. [6] State Key Laboratory of Emerging Infectious Disease, Li Ka Shing Faculty of Medicine, The University of Hong Kong, Hong Kong, China. [7] State Key Laboratory of Oncology in Southern China, Sun Yat-sen University Cancer Center, Guangzhou, China. [8] Department of Pathology, Li Ka Shing Faculty of Medicine, The University of Hong Kong, Hong Kong, China. [9] Department of Pathology, The University of Hong Kong-Shenzhen Hospital, Shenzhen, China. ✉email: xyguan@hku.hk

NPC is a squamous cell carcinoma arising from the nasopharynx epithelium. Although NPC remains uncommon worldwide, it is endemic in East and Southeast Asia and North Africa, indicating the incidence is highly influenced by epidemiological patterns and genetic susceptibility in certain ethical groups[1]. Despite its geographical distribution, the intense immune infiltration and the commonly low degree of differentiation are the unique characteristics that make NPC significantly differ from other cancers[1,2]. Clinical observations have revealed that a substantial amount of stromal cells often intermixes with malignantly transformed nasopharyngeal epithelial cells. Furthermore, the Epstein–Barr virus (EBV) infection is extensively associated with NPC incidence, and significantly shapes the NPC microenvironment via chronic immune activation, so that eventually affects tumor progression and therapeutic outcome[3]. So far, chemoradiotherapy remains the first-line treatment option against NPC, but it often fails to achieve the optimal effect in patients with either advanced or metastatic tumors[4]. Recently, immunotherapy has emerged as a promising strategy in many cancers[5]. For instance, the use of PD-1 inhibitors, such as pembrolizumab and nivolumab, against NPC has generated promising pre-clinical results[6]. Nevertheless, the lack of a comprehensive understanding of the NPC microenvironment constantly hinders the optimization of therapeutic interventions. Therefore, it is essential to further elucidate the landscape, dynamics, and functional characteristics of tumor-infiltrating stromal cells in NPC patients and the nonmalignant counterpart.

Infiltration of dysfunctional and tumor-promoting stromal cells is one of the cancer-dependent hallmarks during malignant transition and progression. The subpopulations of the tumor-associated stromal cells and their interactions are extensively diverse with the evidence for transcriptional alterations and clonal expansion[7]. Previous studies of NPC-associated immune cells have largely focused on EBV-specific chimeric antigen receptor T cells that can induce a high level of antigen-specific cytotoxic response[8,9]. Other TME-residing lymphocytes, myeloid cells, and fibroblasts remain insufficiently identified and characterized in the NPC microenvironment. For example, recent evidence has suggested that B cells constitute an emerging factor in immunotherapeutic outcome for varied malignancies[10–13], but the phenotype and function of infiltrating B cells are primarily unexplored in NPC. Overall, many tumor-infiltrating subpopulations have been increasingly recognized as promising biomarkers and therapeutic targets that can be used in early diagnosis, prognostic prediction and reinvigoration of dysfunctional immunity[14]. However, due to the lack of high throughput single-cell analyzing techniques in the last decade, the stromal heterogeneity and molecular mechanism of stroma-tumor interplay in NPC remain poorly explored. Hence, to comprehensively decipher the complexity of the stromal infiltration and reveal NPC-specific features, we performed 5' single-cell RNA sequencing integrated with V(D)J profiling on 14 patients with either nasopharyngeal carcinoma (NPC) or nasopharyngeal lymphatic hyperplasia (NLH). The in-depth analysis of stromal cells derived from the NPC microenvironment and the non-malignant counterpart revealed the molecular heterogeneity within the TME and the potential effects on tumor progression and treatment outcome.

## Results
**Identification of stromal cells in the NPC microenvironment via single-cell RNA sequencing.** Fourteen patients were enrolled in this study, in which eleven patients were diagnosed with NPC, and the others were diagnosed with NLH. Fresh tissue samples from these patients were collected via endoscopic biopsy. Subsequently, the fresh tissues were immediately dissociated into a single-cell suspension and analyzed by unique transcript counting through barcoding each single cell with a unique molecular identifier (UMI) (Fig. 1a). After human genome mapping and quality filtering, we obtained 66,627 cells where 50,169 cells (75%) originated from the NPC microenvironment and 16,458 cells (25%) from the nonmalignant microenvironment, and detected on average 1215 median genes and 99,739 median reads per cell (Fig. 1b and Supplementary Ffig. S1a). We used the Seurat R package (version 3.0) to perform the down-stream analysis[15]. We applied principal component analysis and a graph-based clustering approach to categorize individual cells into distinct clusters. Then, the Uniform Manifold Approximation and Projection (UMAP) was used to reduce the dimensionality and visualize the cell distribution. To build on this, we assigned the clusters into six major cell lineages via well-recognized marker genes (Fig. 1b, c).

Although myeloid cells and fibroblasts exhibited strong preference in the TME, most of the clusters were not tumor-specific nor patient-specific, indicating that there were no extensive variations in the composition of the major stromal lineages (Fig. 1b, e). The degree of stromal infiltration for each patient was also independently determined by a group of professional pathologists based on hematoxylin and eosin staining and immunohistochemical staining (Fig. 1f and Supplementary Fig. S2a). The interpatient analysis showed that T cells and B cells were the most predominant cell types in the NPC microenvironment, which was consistent with previous findings (Fig. 1d)[16–18]. However, patient 6 with a very rare type of differentiated NPC, showed abnormal myeloid and B cell infiltration, suggesting differentiation grade of NPC might influence the stromal infiltration in the TME. In this study, we comprehensively identified and characterized the heterogeneous subpopulations within the infiltrating T cells, B cells and myeloid cells.

To map the conserved and tumor-specific subpopulations in the infiltrating immune cells, we performed second-round UMAP within the T cells, B cells and myeloid cells. Together, the second-round UMAP more specifically deciphered the heterogeneous components in the TME and non-malignant microenvironment, revealing 36 distinct immune subpopulations. Using the random forest algorithm, we further validated the T and B subpopulations in an independent cohort containing 37,442 single cells obtained from different tissues of patients 9-13, showing a strong clustering correlation between the discovery and validation cohorts (Supplementary Fig. S1b–f). To further define the phenotype, functional state and prognostic value of the major clusters, we performed more in-depth characterization and validation within these genetically distinct subpopulations.

**Identification and characterization of T-cell diversity in the NPC and non-malignant microenvironment.** Fourteen subpopulations were yielded from 32,656 T cells, representing the most prevalent and diverse cell type in the nasopharyngeal microenvironment (Fig. 2a). The expression of marker genes and functional signatures were used to identify and characterize the subpopulations of tumor-infiltrating T cells (Fig. 2d, Supplementary Fig. S3a). For example, three distinct exhausted T cell subpopulations were identified and characterized by *LAG3*, *HAVCR2*, and *TOX*, which were previously reported in T cell dysfunction[19–21]. Although most of T cell subpopulations exhibited minor inter-patient heterogeneity and were primarily shared in malignant and non-malignant microenvironment, the relative abundance and activated/inhibitory signatures were considerably different (Fig. 2c, d). Two subpopulations of exhausted T cells, characterized by high expression of an immune checkpoint *HAVCR2* and an exhaustion-associated transcription

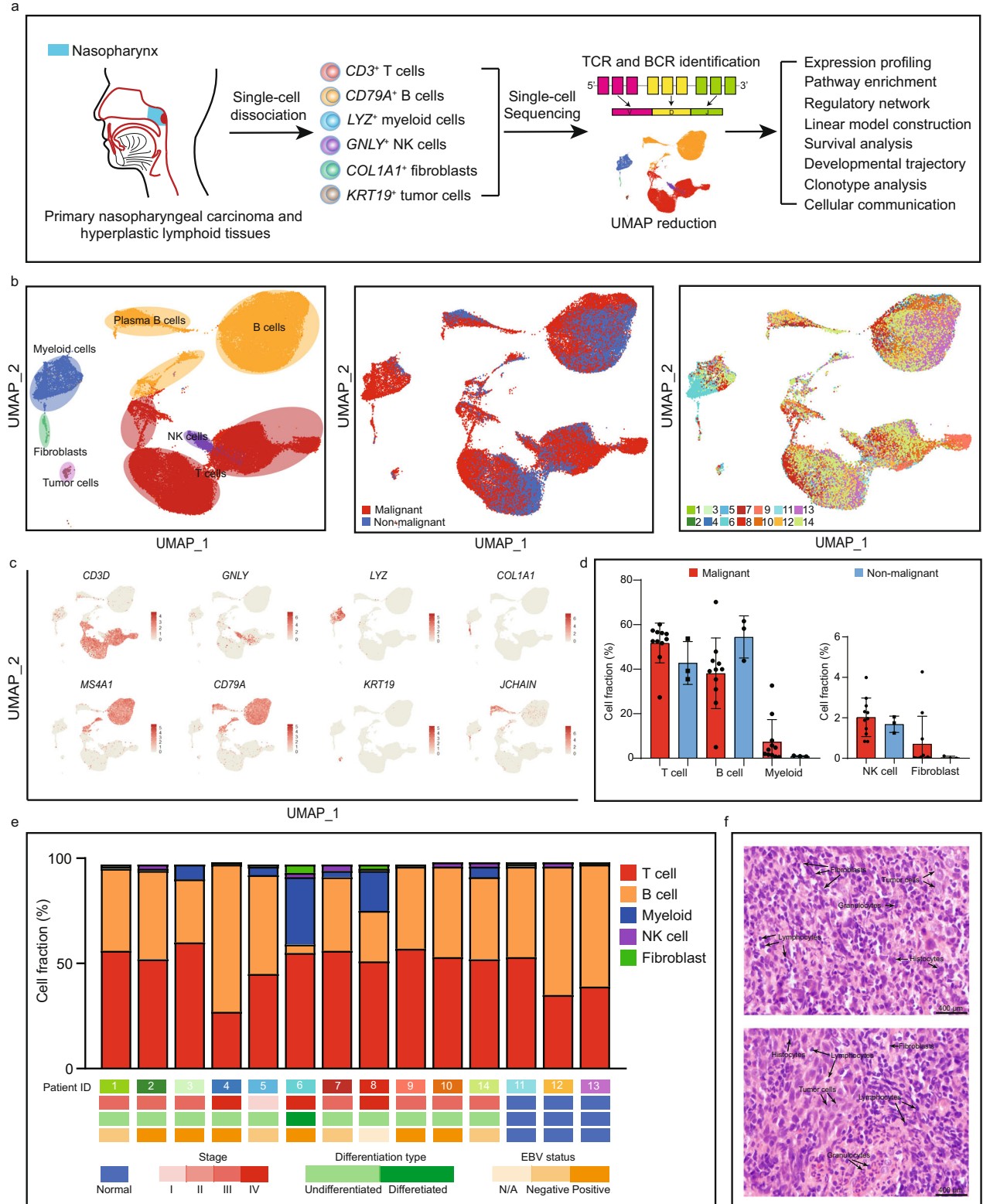

**Fig. 1 The landscape of the microenvironment in nasopharyngeal carcinoma and nasopharyngeal lymphoid hyperplasia at a single-cell resolution.**
**a** Graphical summary of the experimental design. Stromal cells and cancer cells from the NPC and NLH patients were collected and processed by single-cell 5′ RNA sequencing for transcriptomics analysis and coupled V(D)J sequencing for clonal profiling on T and B cells. **b** UMAP plot of 66,627 cells showing the components in the NPC and NLH microenvironment, color-coded by the major cell lineage (left), sample type (middle) and corresponding patient (right). **c** The expression of marker genes for the major lineages defined in **a**. **d** The relative abundance of the major cell lineages in the malignant ($n = 11$ biologically independent samples) and non-malignant ($n = 3$ biologically independent samples) microenvironment. Data are presented as mean values ± SD. **e** The fraction of major cell lineages originating from the 14 patients, with their clinical information. Source data are provided as a Source Data file. **f** The representative H&E staining images (400×) taken from two NPC patients, showing the high stromal infiltration in the NPC microenvironment. Scale bar = 400 μm.

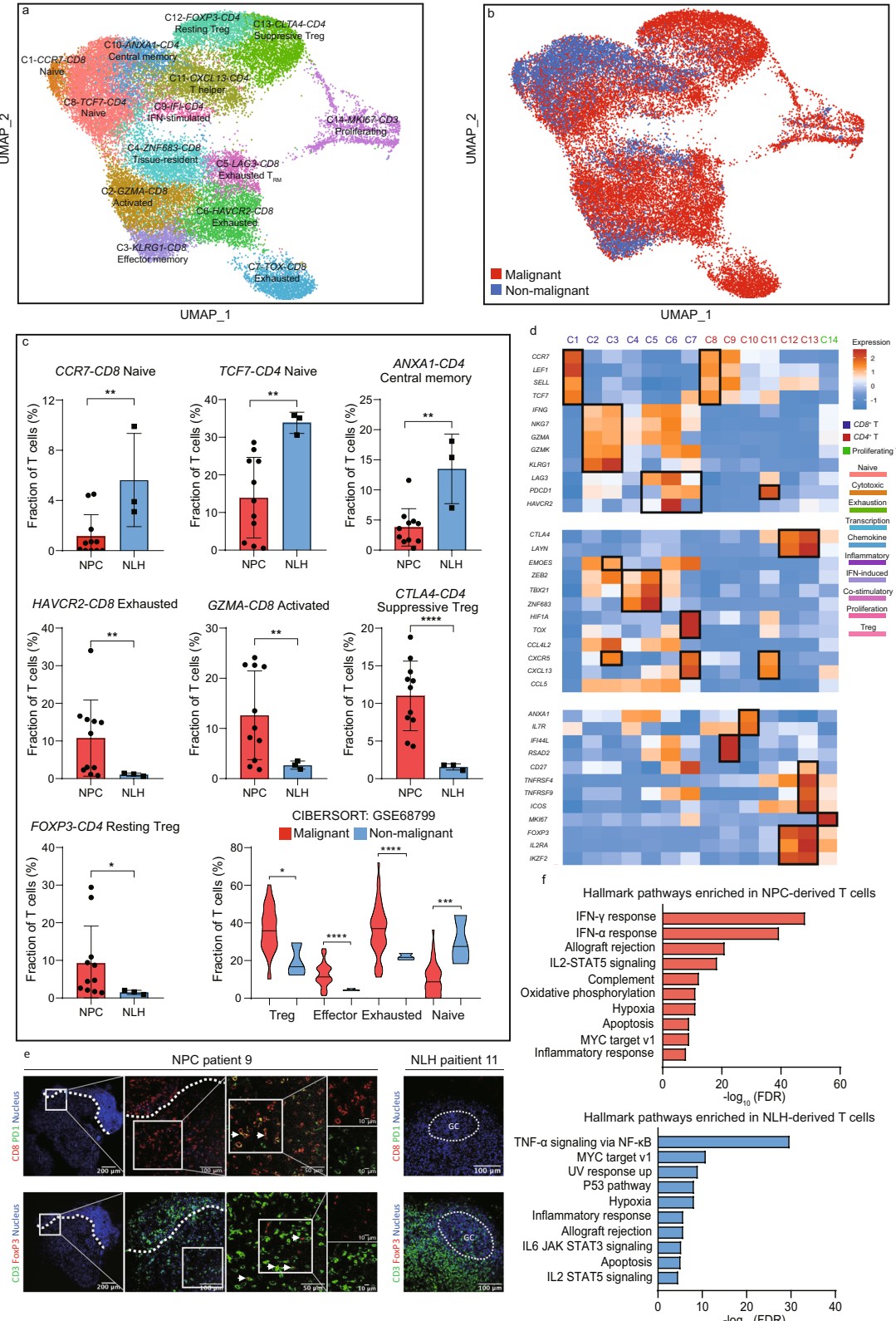

factor *TOX*, respectively, were observed highly enriched in the NPC microenvironment (Fig. 2c). However, unlike *HAVCR2*-exhausted T cells that were constantly distributed in all NPC patients, *TOX*-exhausted T cells were found extensively infiltrated in patient 9, indicating that they were a highly patient-specific

subtype (Supplementary Fig. S3d, e). Other subtypes of T cells, including regulatory T cells and activated T cells, were also preferentially infiltrated in the TME, whereas the naïve T cells and central memory T cells were more likely to reside in the non-malignant microenvironment. Deconvolution analysis via

**Fig. 2 Enrichment of exhausted and immunosuppressive T cell phenotype is a tumor-specific characteristic in the NPC microenvironment. a** UMAP plot of 32,656 T cells from 14 patients, identifying 14 subpopulations. Each dot represents one single cell, color-coded by the cluster. **b** UMAP plot of 32,656 T cells from the NPC microenvironment and NLH microenvironment. Each dot represents one single cell, color-coded by the sample type. **c** The relative abundance of significantly enriched T cell subpopulations in the malignant (For single-cell data, $n = 11$ biologically independent samples; for CIBERSORTx estimation, $n = 42$ biologically independent samples) and non-malignant (For single-cell data, $n = 3$ biologically independent samples; for CIBERSORTx estimation, $n = 3$ biologically independent samples) microenvironment. Each dot represents one patient. The two-sided Student's $t$ test was used to determine the statistical significance. $p$ values $\leq 0.05$ were represented as *, $\leq 0.01$ as **, $\leq 0.001$ as ***, $\leq 0.0001$ as ****. Data are presented as mean values ± SD. Source data are provided as a Source Data file. **d** The normalized average expression of functional signatures for T cells subpopulations defined in **a**. **e** Infiltration of exhausted CD8+ T cells and Treg in the NPC and NLH microenvironment. Representative images of PD1+ CD8+ T cells and FOXP3+ Treg cells in the NPC patient 9 (white arrows). The left photo shows a low magnification overview (×100); the right photo shows the boxed area in the left photo. Representative images of expression of PD1 (FITC) on CD8+ T cells (TRITC, white arrow) and FOXP3 (TRITC) on CD3+ T cells (FITC) in NLH patient 11 (original magnification, ×200). GC, germinal center. Three independent experiments were performed and generated similar results. **f** The GSEA hallmark pathways enriched in the NPC-derived and NLH-derived T cells, ordered by -log10 false discovery rate (FDR).

CIBERSORTx from bulk RNA sequencing data corroborates our finding that effector, exhausted and suppressive T cell subtypes were highly enriched in NPC patients[22] (Fig. 2c). We further validated the enrichment of exhausted T cells and Tregs via multicolor immunofluorescence of CD3, CD8, FOXP3, and PD-1, showing a consistent result with our single-cell data (Fig. 2e). These findings suggested that the severe infiltration of dysfunctional and immunosuppressive T cells largely affected T-cell immunity against NPC. Hence, we later compared the hallmark pathways in NPC and non-malignant microenvironment-derived T cells. The pathway analysis revealed that interferon-gamma (IFN-γ) response and interferon-alpha (IFN-α) response were upregulated in the NPC-derived T cells, suggesting that the excessive production of type 1 and type 2 interferons (IFNs) due to chronic activation might significantly influence the composition and functional state of the infiltrating T cells (Fig. 2f). The T cells derived from the non-malignant microenvironment were mostly influenced by the nuclear factor kappa-light-chain-enhancer of activated B cells (NF-κB) pathway (Fig. 2f). Subsequently, we performed z-normalized pathway analysis and identified several major pathways that were differently enriched across the *CD8+* and *CD4+* T cell subpopulations (Supplementary Fig. S3b). For instance, exhausted T cells showed loss of IFN responses that might explain its dysfunctional state, whereas suppressive Tregs showed increased IFN responses compared with resting Tregs, indicating that IFN-associated pathways might lead to Treg-associated immune suppression. Besides, the apoptosis and P53 pathways were found significantly downregulated in exhausted T cells and suppressive Tregs that might facilitate their proliferation in the TME. Other pathways, such as glycolysis, allograft rejection, and oxidative phosphorylation showed a high preference in the TME-resided T cells, suggesting that targeting these pathways might enhance immunotherapeutic efficiency. Overall, the NPC-derived T cells exhibited apparent functional differences compared to the NLH-derived counterpart.

To further corroborate the molecular variations between T cell infiltrates in the NPC and non-malignant microenvironment. We performed MAST analysis to identify differentially expressed genes (DEGs) in T cells (Fig. 3a). Among all the DEGs, we predicted upstream transcription factors of the top 30 upregulated genes via RegNetwork and constructed the gene regulatory network via Cytoscape (Fig. 3b). Intriguingly, we found that *CXCL13* and *LGALS1* were significantly upregulated in the NPC-derived T cells. *CXCL13* was one of the gene signatures for follicular T helper cells, but it was also found highly expressed on exhausted T cells, particularly *PD-1+* T cells (Fig. 3c). Previous studies had also reported that *CXCL13* was highly expressed in PD-1^High exhausted T cells, and it was probably involved in the formation of tertiary lymphoid structures (TLSs) via *CXCR5+* B cell recruitment[23,24]. Therefore, we divided the NPC patients into

*CXCL13*-high and *CXCL13*-low groups based on the median gene expression, and observed that the fraction of the *CXCR5+* B cells was significantly higher in *CXCL13*-high patients (Fig. 3e). In addition, higher expression of *CXCL13* was found correlated to better progression-free survival in NPC patients, suggesting the *CXCL13*-high exhausted T cells might remain beneficial to immune modulation (Fig. 3f).

*LGALS1* has been reported to have the immunomodulatory function in varied immune cells[25,26]. In this study, *LGALS1* was found highly expressed in NPC-associated Tregs (Fig. 3d). We also discovered that the fraction of immunosuppressive Tregs, instead of resting Tregs, were significantly higher in *LGALS1*-high patients, indicating it might play a vital role in Treg activation (Fig. 3e). A previous study has shown that galectin-1–homozygous null mutant mice exhibited impeded Treg activity, confirming its immunosuppressive role[27]. Although *LGALS1* might serve as a promising therapeutic target to revert the immunosuppressive function in Tregs, the molecular mechanism of how *LGALS1* deficiency affects Treg activity remains largely unexplored. Based on the gene regulatory network, *S100A11* and *S100A4* were found upregulated in *LGALS1+* Treg and predicted to interact with *LGALS1*, suggesting its regulatory effect might be exerted via a calcium channel-dependent process (Fig. 3b).

**Functional modules, clonal analysis and developmental trajectory revealed the dynamics of exhausted T cells and Tregs in the NPC microenvironment.** The down-stream analysis revealed a rich set of functional modules that were associated with naïve, cytotoxic, exhausted and immunoregulatory programs in the T cell subpopulations. Thus, we identified the most representative genes in these programs, including *CCR7* (naïve), *NKG7* (cytotoxic), *LAG3* (exhaustion), and *FOXP3* (Treg). We later constructed gene profiles containing the top ten genes that were mostly correlated to the representative genes (Fig. 3g–j). Based on these gene profiles, we subsequently established generalized linear models in naïve T cells, cytotoxic T cells, exhausted T cells and Tregs using the glm function in R. The generalized linear models assigned a distinct parameter to each gene in the profile based on its expression weight. Subsequently, the functional scores were calculated based on the parameters estimated from the model, which represented the relative abundance of certain cell types.

We firstly performed the quantitative comparison of the cytotoxic program within the *CD8+* T cell clusters. The result demonstrated that the exhausted T cell clusters (C5, C6, and C7) maintained intermediate-to-high cytotoxic activity instead of being terminally dysfunctional, suggesting there might be a transitional process where effector T cells gradually lost cytotoxicity due to chronic activation and tumor-dependent mechanisms in the TME (Fig. 3g). Also, the analysis of the exhaustion program showed that exhaustion was not only

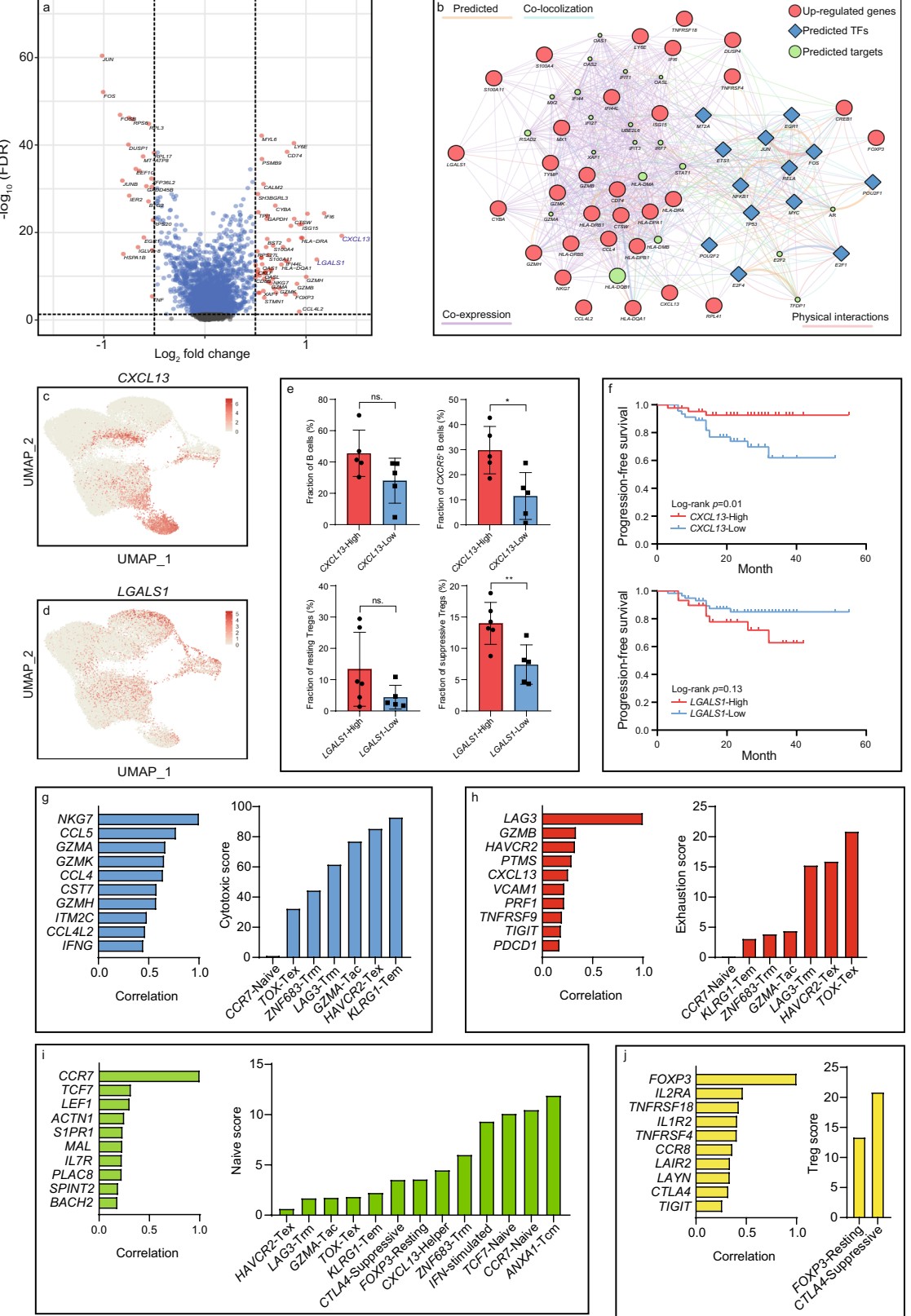

induced in the exhausted T cells, but also induced in the effector T cells, but at a lower degree, stating that the activation-to-exhaustion transition was indeed a dynamic process where T cells could be in either a progenitor, intermediate or late exhausted state (Fig. 3h). Consistent with the clustering, the quantitative analysis of naïve program across all T cells showed that $CD4^+$

and $CD8^+$ naïve T cells had a high naïve score (Fig. 3i). Analysis of the Treg program suggested that suppressive Tregs possessed more potent immunoregulatory capacity than the resting Tregs (Fig. 3j). Such increased capacity could be explained by the higher expression of stimulatory molecules-encoding genes in the suppressive Tregs, including *CD27*, *TNFRSF4*, *TNFRSF9*,

**Fig. 3 Identification of NPC-specific T cell signatures and characterization of their correlations to patient prognosis and T cell dynamics. a** The differentially expressed genes (log$_2$ fold change $\geq$0.5, FDR $\leq$ 5 × 10$^{-2}$) in NPC-derived and NLH-derived T cells identified by the MAST analysis. **b** The gene regulatory network constructed by Cytoscape, colored for the associated gene type (red: top 30 upregulated genes from NPC-derived T cells, blue: the upstream transcription factors predicted by RegNetwork, green: the targeted genes predicted by GeneMANIA). **c** The expression of *CXCL13* in each T cell profiled on the UMAP. **d** The expression of *LGALS1* in each T cell profiled on the UMAP. **e** The total B cell fraction and *CXCR5*$^+$ B cell fraction in *CXCL13*-high ($n$ = 5 biologically independent samples) and CXCL13-low groups ($n$ = 5 biologically independent samples); The resting Treg fraction and suppressive Treg faction in *LGALS1*-high ($n$ = 6 biologically independent samples) and *LGALS1*-low groups ($n$ = 5 biologically independent samples). Each dot represents one patient. The two-sided Student's t-test was used to determine the statistical significance, *p*-values >0.05 were represented as ns. (not significant), *p* values $\leq$ 0.05 were represented as *, $\leq$0.01 as **. Data are presented as mean values ± SD. **f** The progression-free survival for 88 NPC patients, stratified for the normalized average expression (binary: high expression vs. low expression) of *CXCL13* and *LGALS1*. *p* values were determined based on the two-sided log-rank test. **g** The most correlated genes to *NKG7* (the representative gene in the cytotoxic module) across cytotoxic T cells and predicted cytotoxic scores for *CD8*$^+$ subpopulations based on the linear model constructed by this set of genes. **h** Similar to **g** but showing the most correlated genes to *LAG3* (the representative gene in the exhaustion module) across exhausted T cells and predicted exhaustion scores for *CD8*$^+$ subpopulations. **i** The most correlated genes to *CCR7* (the representative gene in the naïve module) across naïve T cells and predicted naïve scores for all T cell subpopulations. **j** The most correlated genes to *FOXP3* (the representative gene in the Treg module) across Tregs and predicted Treg scores for Treg subpopulations. Source data are provided as a Source Data file.

*TNFRSF18,* and *ICOS*. To validate the reliability of our linear models, we performed the correlation analysis between functional scores and corresponding abundances of T cell subpopulations in the discovery cohort and validation cohort. The result exhibited that functional scores were highly correlated to the T cell abundance, suggesting the linear model as an alternative approach to comprehensively and accurately estimate dynamic functional states of T cells in the NPC microenvironment (Fig. 4a). In addition, we analyzed the T cell subpopulations from an independent NPC single-cell cohort done by Chen et al.[28] The functional scores calculated by our model in Chen's data also showed high correlations to the abundance of selected T cell subpopulations, which further demonstrated the reliability of our linear model (Fig. 4b). Meanwhile, we computed the correlation between the expression of representative genes and functional scores in two independent bulk RNA sequencing cohorts, containing 45 patients and 88 patients, respectively[29]. The functional scores were significantly correlated to the corresponding genes, indicating that the linear model is capable of more comprehensively and accurately quantifying real dynamic T cell functional states from bulk RNA sequencing data, compared to the signature-based characterization. (Fig. 4c). We later linked the functional scores of 88 NPC patients with their clinical information and found higher naïve score was significantly correlated to better progression-free survival (Supplementary Fig. S3f).

To characterize the clonal diversity of infiltrated T cells in the NPC and non-malignant microenvironment, we systematically analyzed the identity of TCRs on individual T cells obtained from the coupled V(D)J profiling. In total, we detected on average 2,071 unique clonotypes per patient. As expected, the clonotypes were highly variable across the patients, but the composition of clonal sizes remained relatively consistent in NPC and NLH patients. Therefore, we divided the clonotypes into three categories based on the number of cells assigned to each clone (refer to clonal size $\geq$3, =2 and =1) (Fig. 5a). The fraction of larger clone sizes (clonal size $\geq$3 and =2) were found significantly enriched in the NPC patients, indicating a strong T-cell clonal expansion in the TME (Fig. 5b). In contrast, the T-cell expansion in the non-malignant microenvironment was relatively static since it was largely dominated by T cells with small clonality (clonal size = 1) (Fig. 5b). We later computed the exhaustion and Treg scores of patients 4–13, and found that both scores were positively correlated to large clonality. This result illustrated that the exhausted T cells and Tregs in the TME underwent a rapid clonal expansion and eventually resulted in a higher abundance of these T cell subtypes in NPC patients (Fig. 5c).

To further understand the clonal expansion and its influence on T-cell dynamics, the top three largest clones in each patient were projected on the UMAP with their corresponding clonotypes (Fig. 5d and Supplementary Fig. S4a, b). The activated (C2 and C3) and exhausted T cells (C5, C6, and C7) were shared identical clonotypes in most of the NPC patients, validating the presence of activation-to-exhaustion transition. Nevertheless, we found a substantial amount of *TOX*-exhausted T cells were only shared identical clonotypes with *HAVCR2*-exhausted T cells. The result inferred that the exhausted T cells in different clusters were not completely independent, and the state transition could happen from one exhausted subtype to another, namely from *HAVCR2*-exhausted to *TOX*-exhausted or vice versa. It attracted our attention that a significant fraction of proliferative T cells also shared identical clonotypes with the exhausted T cells, indicating the exhausted program was often accompanied by elevated proliferation. Besides, we conducted TCR motif analysis and found that although most of the motifs were consistent in T cells either derived from the NPC or NLH microenvironment, there still existed a small portion of TCR patterns that were NPC-specific (Supplementary Fig. S4c). Taken together, it was noteworthy to mention that C5-*LAG3*, C6-*HAVCR2* and C7-*TOX* exhausted T cells with different transcriptional states exhibited distinct cytotoxic and exhausted activities. It demonstrated that the exhausted T cells that had been previously reported as a relatively homogenous population, were possibly highly heterogeneous and could be originated from varied sources, including activation-to-exhaustion transition and clonal expansion.

The pseudotime developmental trajectory was independently performed on *CD8*$^+$ T cells and *CD4*$^+$ T cells via Monocle v2[30,31]. The developmental trajectory provided a unique visualization that inferred the lineage structure of T cells in the NPC and non-malignant microenvironment. The trajectory *CD8*$^+$ T cells showed that the *HAVCR2*-exhausted, *TOX*-exhausted, *LAG3*-exhausted T cells, and $T_{RM}$ were located at the end of different branches, indicating their distinct developmental processes (Fig. 5e and Supplementary Fig. S3c). The effector and memory effector T cells were positioned in between, suggesting that they were at an intermediate developmental state. The result further demonstrated that exhaustion was the terminal stage during *CD8*$^+$ T cells development, and reprogramming exhausted T cells via TME modulation might become a feasible therapeutics to re-activate anti-tumor immunity. The trajectory *CD4*$^+$ T cells also exhibited the transitional process from naïve T cells to Tregs, central memory T cells, and T follicular helper cells (Fig. 5f and Supplementary Fig. S3c). The pseudotime

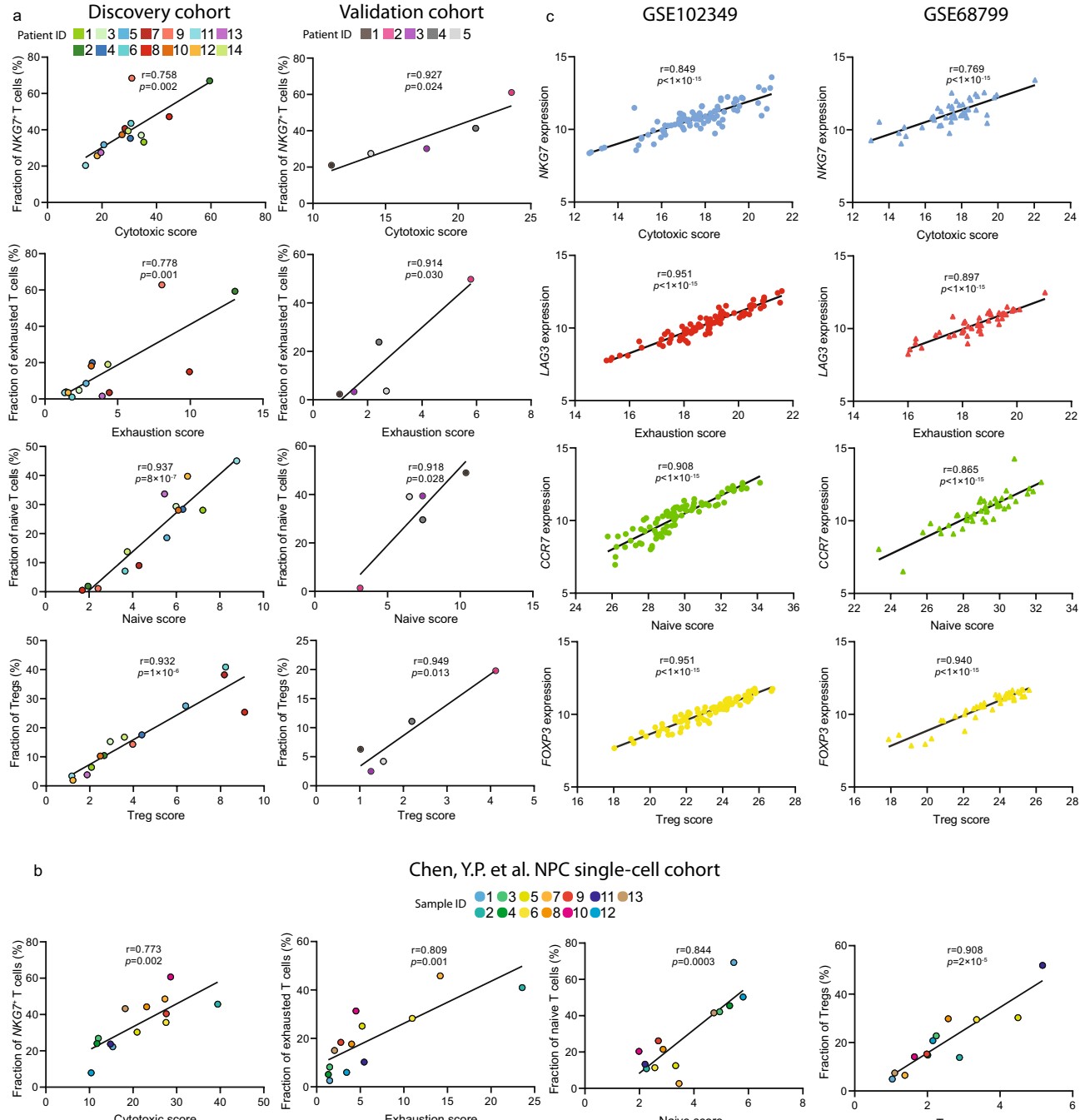

**Fig. 4 Independent validation of the functional modules in bulk RNA sequencing and single-cell RNA sequencing cohorts. a** The Pearson correlation analysis of functional scores (cytotoxic, exhaustion, naïve, and Treg) and the abundance of T cell subpopulations (*NKG7*+ cytotoxic T cells, exhausted T cells, naïve T cells, and Tregs) in the discovery cohort and validation cohort. **b** The Pearson's correlation analysis of functional scores (cytotoxic, exhaustion, naïve, and Treg) and the abundance of T cell subpopulations (*NKG7*+ cytotoxic T cells, exhausted T cells, naïve T cells, and Tregs) in PY Chen, et al. NPC single-cell cohort. **c** The Pearson correlation analysis of functional scores and the corresponding genes in two GEO cohorts (GSE102349 and GSE68799), respectively. The two-sided *p* value for each functional module in GSE102349 and GSE68799 was extremely small (<1 × 10⁻¹⁵), therefore the exact *p* values cannot be shown in the figure. Source data are provided as a Source Data file.

estimation showed that the resting Tregs were considered as an earlier developed subtype, which further supported our previous finding that the suppressive Tregs, as a more mature form, had the more potent immunoregulatory capability and underwent more robust clonal expansion.

**Deciphering the prognostic value of B cell-specific signatures and NPC-associated functions of B cell subtypes.** B cells were

previously considered as a relatively homogenous population in the TME. Nevertheless, we detected 27,506 B cells in 11 distinct subtypes, representing the second-largest and second-most diverse stromal cell type in the TME (Fig. 6a, b and Supplementary Fig. S5a). The unique clusters of innate-like and germinal center B cells were first to be found in the intratumor mass of NPC. Plasma B cells were specifically enriched in patient 2, patient 6 and patient 7 and primarily contributed to the inter-

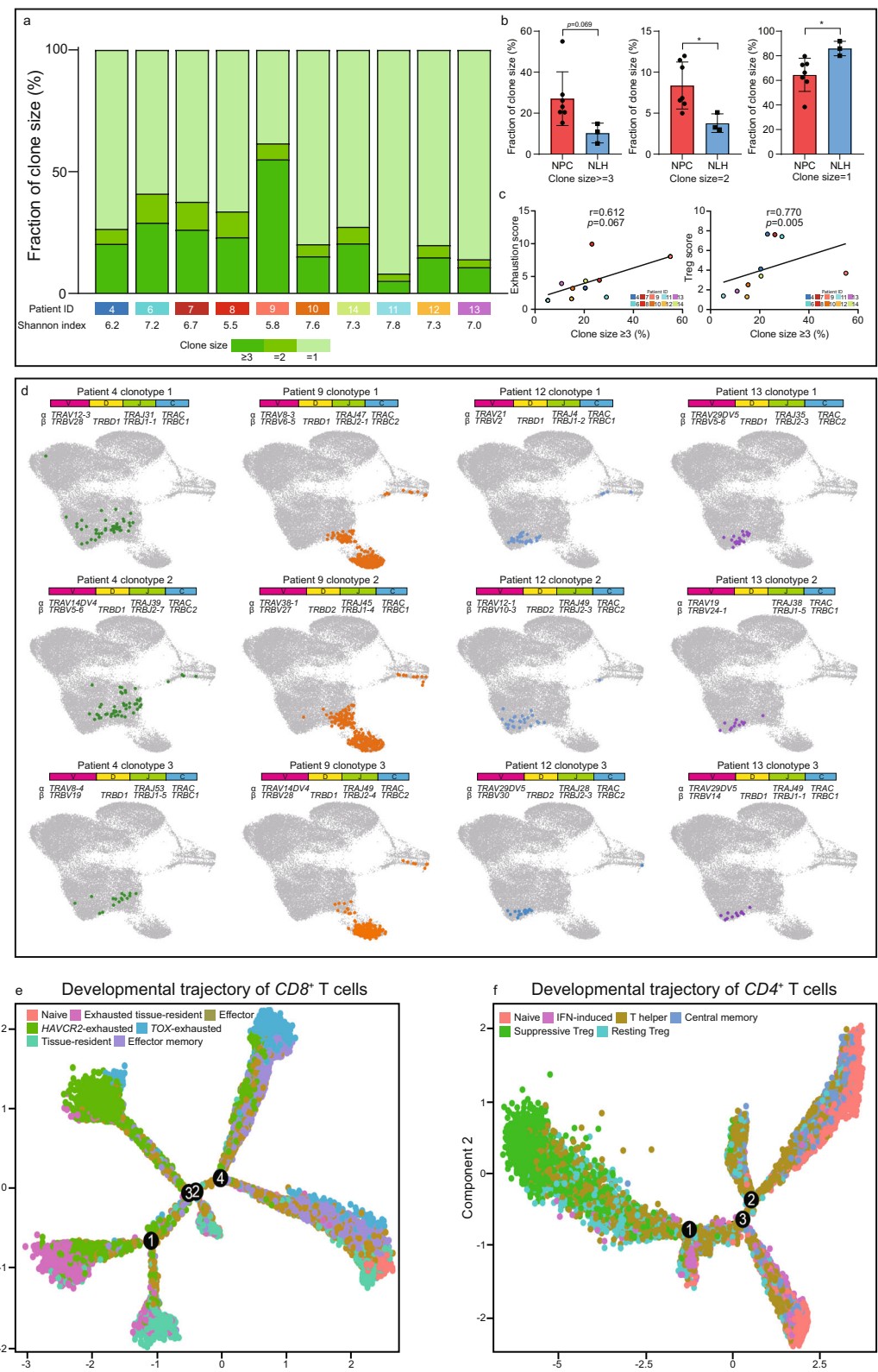

**Fig. 5 Single-cell V(D)J sequencing reveals the clonal expansion and activation-to-exhaustion transition in exhausted and immunosuppressive T cell subtypes. a** Distribution of T cell clones by size, with different Shannon index showing the clonal diversity for each patient. **b** The relative abundance of different clone sizes in the NPC-derived (*n* = 7 biologically independent samples) and NLH-derived (*n* = 3 biologically independent samples) T cells. Each dot represents one patient. The two-sided Student's *t* test was used to determine the statistical significance, *p* values ≤ 0.05 were represented as *, ≤0.01 as **, ≤0.001 as ***, ≤0.0001 as ****. Data are presented as mean values ± SD. **c** The Spearman's correlation between the calculated exhaustion score/ Treg score and the fraction of clones with ≥3 cells in patients. Source data are provided as a Source Data file. **d** UMAP plot of the top three largest clones in patients 4, 9 (NPC) and 12, 13 (NLH), with their associated clonotypes. (**e** and **f**) The developmental trajectory of *CD8*+ and *CD4*+ T cells, colored-coded by the associated cell subpopulations and numbered by different developmental branches.

patient heterogeneity of B cells in the NPC microenvironment (Fig. 6c and supplementary fig. 5b). Although the number of plasma B cells and switched memory B cells was significantly increased in the TME, it did not reach statistical significance due to considerable interpatient variation (Fig. 6c and Supplementary Fig. S5c). Total naïve B cells and innate-like B cells were found enriched in the non-malignant microenvironment, whereas the IFN-induced B cells (including the naïve form) and double-negative B cells were particularly NPC-enriched (Fig. 6d). From the bulk RNA sequencing data, deconvolution analysis validated that plasma B cells and double-negative B cells were more likely to infiltrate in the NPC microenvironment. Whereas pan B cells, including naïve B cells, unactivated B cells and innate-like B cells, were preferentially enriched in the nonmalignant microenvironment (Fig. 6e). The genetic differences between NPC-derived and NLH-derived B cells were rare, suggesting that the tumor-associated function of B cells might largely depend on the abundance instead of the differences in genetic regulations or cytokine secretion. The differentially expressed genes in NPC-derived B cells were primarily interferon-induced and immunoglobins-encoding genes, including *IFITM1*, *IFI44L*, *IGHA1*, *IGKC*, and *IGHG3* (Supplementary Fig. S5d). Consistent with the results shown in T cell subpopulations, the pathway analysis of B cells revealed that the NPC-derived B cells were also largely influenced by the excessive release of IFN-γ and IFN-α, whereas the NLH-derived B cells were altered due to inflammatory disorder (Fig. 6f). Previous studies have reported that the effects of IFNs on B cells were associated with the enhancement of maturation, differentiation, immune activation and survival[32,33]. However, the function of those IFN-induced B cells in tumor progression was mainly unexplored. Emerging evidence has suggested that IFNs might trigger immunosuppressive mechanisms[34], but whether the enrichment of IFN-induced B cells was prognostic in NPC patients remains unclear. The double-negative B cells were previously found expanded in non-small-cell lung carcinoma (NSCLC) and negatively correlated to the number of activated B cells, which was prognostic for better overall survival in NSCLC patients[35]. Since the presence of tumor-infiltrating B cells was usually correlated to better overall survival and treatment outcomes, we selected several representative B-cell signatures (*CD79A*, *MS4A1*, *IGHD*, and *FCRL4*) and examined their prognostic values in NPC patients. The survival analysis revealed that the signature genes were significantly correlated to better progression-free survival in NPC patients, suggesting an active role for B cells in anti-NPC immunity (Supplementary Fig. S5e).

To further gain insights into the function of infiltrating B cells, we performed the clonal analysis to quantify the distribution and diversity of clonotypes in each patient (Fig. 6g). In total, we detected on average 2419 unique clonotypes per patient. Similar to the result of T cells, larger B cell clones were significantly enriched in NPC patients, whereas the smaller clones were more abundant in NLH patients (Fig. 6h). We later examined the frequently enriched B-cell clonotypes in NPC and NLH patients, respectively (Supplementary Fig. S6a). We conducted the motif analysis and found that all of the BCR motifs were highly consistent in the composition but remained distinct in frequency (Supplementary fig. 6b). We subsequently performed the developmental trajectory of B cells and revealed the developmental lineage from naïve B cells to different types of mature B cells, including plasma B cells, switched memory B cells and IFN-induced B cells. We also found that many B cell subtypes were at the intermediate development state, such as unactivated B cells, germinal center B cells, and double-negative B cells (Supplementary Fig. S6c).

**Heterogeneity and dynamics of myeloid subpopulations in the NPC microenvironment**. With 3671 myeloid cells clustered into 11 subpopulations, we found that 95% of the cells were derived from NPC patients, indicating the myeloid accumulation was an NPC-dependent process (Fig. 7a, b). Our single-cell data showed that there existed three major myeloid-lineage components, including myeloid-derived suppressor cells (MDSCs), macrophages and dendritic cells (DCs) (Fig. 7c). Previous studies have also reported that the MDSCs, macrophages and DCs were the most abundant myeloid-lineage subtypes in the NPC microenvironment[36–38], and we further classified these cells into finer subpopulations (Fig. 7a, d). Compared to T cell and B cell subpopulations, the myeloid cells, especially TAMs, displayed extensive heterogeneity driven by interpatient variation and tissue specificity (Fig. 7e and Supplementary Fig. S7a). For instance, TAMs primarily derived from patient 6 (C1, C2, and C3) and patient 8 (C4) exhibited highly distinct genetic profiles (Fig. 7c and Supplementary Fig. S7a). In addition, deconvolution analysis revealed that MDSC were highly enriched in NPC patients (Fig. 6f). DCs were found not only resided in the TME, but also frequently infiltrated in the non-malignant microenvironment so that did not yield a statistical significance. Pathway enrichment of NPC-derived and NLH-derived myeloid cells confirmed that the presence of IFNs also severely shaped the myeloid landscape in the NPC patients (Fig. 7g). The MAST analysis and gene regulatory network revealed the co-expression of M2 and M1 polarization-associated genes in the TAMs, including *FCGR3A*, *FCGR2A*, *TREM2,* and *APOC1*, indicating that the TAM subpopulations might be more complicated than the conventional M1/M2 classification (Supplementary Fig. S7b, c)[39,40]. The developmental trajectory validated the dynamics of myeloid-lineage cells. We found C7-*LAMP3*, C8-*IDO1,* and C9-*LGALS2* DCs were grouped on branch 4, indicating that they were developmentally similar (Supplementary Fig. S7d). However, C6-Langerhans cells, a type of specialized tissue-resident DCs, were distinctly grouped on branch 3. Macrophages were mainly located at the end of branches 1 and 2. C1-*FOLR2* and C2-*CCL20* TAMs exhibited distinct developmental programs, and C3-*FCGR2B* TAMs were mainly located in the middle, indicating its intermediate developmental state. MDSCs, including C10-*S100A9* and C11-*NFKB1*, were primarily located on the end of branch 4, which suggested that they were developmental similar but transcriptionally different subtypes.

**Fibroblasts and NK cells as microenvironment-based targets in NPC treatment**. Fibroblasts and NK cells represented two kinds of minor subpopulations in the NPC microenvironment[18,41,42]. Unlike NK cells that were found in both TME and non-malignant microenvironment, fibroblasts were overwhelmingly enriched in NPC patients (Fig. 1d, e). MAST analysis revealed the distinct gene expression profiles in the fibroblasts and NK cells (Supplementary Fig. S7e). Genes that encoded extracellular matrix (ECM) components, such as *COL1A1*, *COL1A2*, *LUM* and *FN1*, were specifically expressed by fibroblasts, suggesting that the ECM in the NPC microenvironment was highly complex and might interact with surrounding tumor and stromal cells via integrin signaling. Thus, inhibition of particular integrin receptors on tumor and immune cells might serve as a potential therapeutic target to disrupt ECM-dependent tumor progression and suppressive immunomodulation. Similar to effector T cells, NK cells commonly expressed a high level of cytotoxic genes, including NKG7, *GZMA*, *GZMB,* and *GZMH*. However, *GNLY* and *KLRD1* were preferentially expressed by NK cells (Supplementary Fig. S7e). Recent

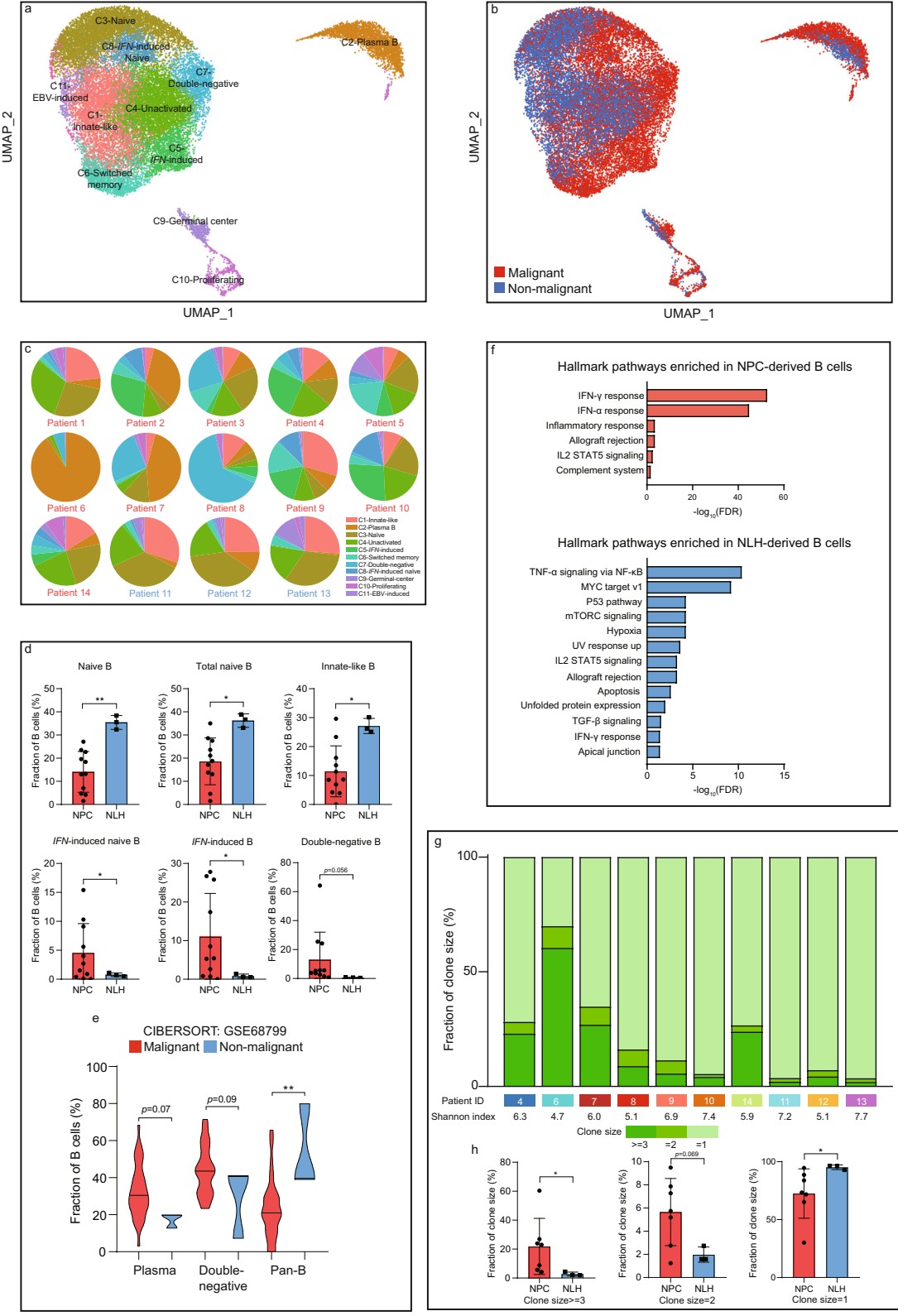

advances suggested that NK cells could also become exhibited after chronic activation, characterized by the upregulation of *KIR*, *NKG2A*, *LAG3*, and *HAVCR2*[43,44], but we did not observe the up-regulation of these exhaustion markers in NK cells. Furthermore, NK cells differentially expressed chemokine-encoding genes *XCL1* and *XCL2*, which were previously reported to recruit *CCR5*[+] DCs. Thus, NK cells in the NPC microenvironment remained immune activated and might serve as a positive regulator in the immune response against NPC.

**Correlation between immune dynamics and patient survival, and cell–cell communications in the infiltrating immune cells.** Based on our single-cell sequencing data, we generated an

**Fig. 6 Clonal expansion and enrichment of immune-activated and IFN-associated B cells is a tumor-specific characteristic in the NPC microenvironment. a** UMAP plot of 27,506 B cells, color-coded by the associated cluster. **b** UMAP plot of 27,506 B cells, color-coded by the sample type. **c** The inter-patient distribution of B cell subpopulations, shown by the percentage of total B cells. **d** The relative abundance of significantly enriched subpopulations in the NPC (n = 11 biologically independent samples) and NLH (n = 3 biologically independent samples) microenvironment. Each dot represents one patient. The two-sided Student's t test was used to determine the statistical significance, p values ≤0.05 were represented as *, ≤0.01 as **. Data are presented as mean values ±SD. **e** The estimated abundance of selected B cell subpopulations via CIBERSORTx in the NPC patients (n = 42 biologically independent samples) and normal patients (n = 3 biologically independent samples). The two-sided Student's t test was used to determine the statistical significance, p values ≤ 0.05 were represented as *, ≤0.01 as **. **f** The GSEA hallmark pathways enriched in NPC-derived and NLH-derived B cells, ordered by -log$_{10}$ FDR. (g) Distribution of B cell clones by size, with different Shannon index showing the clonal diversity for each patient. **h** The relative abundance of different clone sizes in the NPC-derived (n = 7 biologically independent samples) and NLH-derived (n = 3 biologically independent samples) B cells. Each dot represents one patient. The two-sided Student's t test was used to determine the statistical significance, p values ≤ 0.05 were represented as *. Data are presented as mean values ± SD. Source data are provided as a Source Data file.

expression matrix containing 733 representative genes for selected major subpopulations in T cells, B cells and myeloid cells, respectively. By using the annotated signature matrix, the RNA-sequencing data of 88 NPC patients was deconvoluted to estimate the immune abundance via CIBERSORTx (Fig. 8a). The patients were further clustered into three groups based on immune abundance, with either activated tumor immune microenvironment (TIME, cluster 1 and 2) or inactivated TIME (cluster 3). The functional score for each patient was also calculated and normalized, which was later used to characterize the dynamic status of T cells (Fig. 8b). Subsequently, we evaluated the hazard ratio for each immune subtype and patient cluster. The result exhibited that exhausted T cells were correlated with better progression-free survival, whereas other T cell subtypes did not yield a significant correlation with survival (Fig. 8c). Moreover, a higher abundance of plasma B cells, DCs and macrophages was also associated with a better prognosis. Consistent with the previous studies, a higher percentage of double-negative B cells and MDSCs in NPC patients was predictive of worse progression-free survival[35,45]. Thus, targeting double-negative B cells and MDSCs might serve as a promising therapeutic approach in NPC, but further investigations are needed to fully understand the molecular mechanism of these subtypes in the NPC microenvironment. In addition, patients with low T-immune scores or inactivated TIME were shown to have a worse prognosis. Currently, clinical models for patient stratification, survival prediction and treatment evaluation in NPC remain lacking. The single-cell-based devolution method incorporated with functional modules enhanced its reliability as an integrated and feasible model to classify disease, predict survival and therapeutic outcome, but it remains necessary to validate our findings in larger clinical cohorts.

To depict the differences in molecular interaction between the cells derived from the NPC microenvironment and those from the NLH microenvironment, we applied CellPhoneDB to construct a cell–cell communication network via known ligand-receptor pairs within the 38 identified subpopulations[46]. Since T cells and B cells were highly infiltrated in the microenvironment, we visualized the cellular communications within the major T and B subpopulations (Fig. 9a). The result showed that three clusters of exhausted T cells (*HAVCR2*, *TOX*, and *LAG3*) in the TME preferentially interacted with memory B cells, innate-like B cells, unactivated B cells, IFN-induced B cells, but not with plasma B cells, naïve B cells and double-negative B cells, due to lack of interacting pairs in these subtypes. Furthermore, we identified specific ligand-receptor pairs that were constantly involved in T cell and B cell communication. We found that resting Tregs, compared to suppressive Tregs, had much fewer interacting pairs with exhausted T cells, indicating that they were a less dynamic and immunosuppressive subtype (Fig. 9b). Consistent with the findings above, the *HAVCR2* and *TOX*

exhausted T cells in the TME exhibited a strong interaction with varied B cell subpopulations, primarily via the *CXCL13–CXCR5* axis (Fig. 9c). Notably, the NPC-derived exhausted T cells and suppressive Tregs possessed a higher number of ligand-receptor pairs, whereas the resting Tregs, double-negative B cells and tissue-resident T cells possessed fewer pairs, compared to their non-malignant counterparts (Fig. 9d).

## Discussion

Together, we comprehensively deciphered the stromal composition of the NPC microenvironment at single-cell resolution. More importantly, our in-depth analysis has successfully revealed phenotypic and molecular differences between stromal cells derived from the TME and those from the nonmalignant microenvironment. The results exhibited a substantial number of unique features in the NPC microenvironment that had not been previously reported, and highlight key characteristics for further implications in TME-targeted therapy and immunotherapy. Importantly, the NPC-specific characteristics and their relationship with patient stratification and survival significantly contribute to the development of personalized diagnostics and therapeutics. It can also serve as a public-available cohort where hypothesis from other cancers, especially EBV-associated malignancies, can be independently validated, and used to explore promising therapeutic targets and prognostic biomarkers. Our findings were carefully examined in our single-cell validation cohort, NPC single-cell cohort from another group, the two NPC bulk RNA sequencing cohorts, and by staining of formalin-Fixed paraffin-embedded tissues. Additionally, our findings, such as stromal compositions and immune signatures, are highly consistent with the results from recent single-cell studies on NPC, indicating the reliability and reproducibility of our single-cell data[28,47].

In the present study, we have proven that the NPC microenvironment is indeed more complex and heterogeneous than previously reported. In total, we analyzed 104,069 cells from 14 patients (66,627 in the detecting cohort and 37,442 in the validation cohort) and revealed 38 distinct stromal populations, including fibroblasts, NK cells and different subtypes of T cells, B cells and myeloid cells. Although each cell subpopulation exhibited divergent pathway activities, IFN-γ and IFN-α responses profoundly influenced the composition and function of the TME-residing cells. Nevertheless, the molecular mechanism of IFN-associated alterations remains largely unexplored. Further understanding of the IFN- and immune-enriched NPC microenvironment might fuel the advances in treatment strategies against NPC and other immunologically activated cancers.

Subsequently, many NPC-specific signatures were found correlated to patient survival, indicating that the relative abundance of various stromal subpopulations and immune activation status in NPC patients directly influenced tumor development and

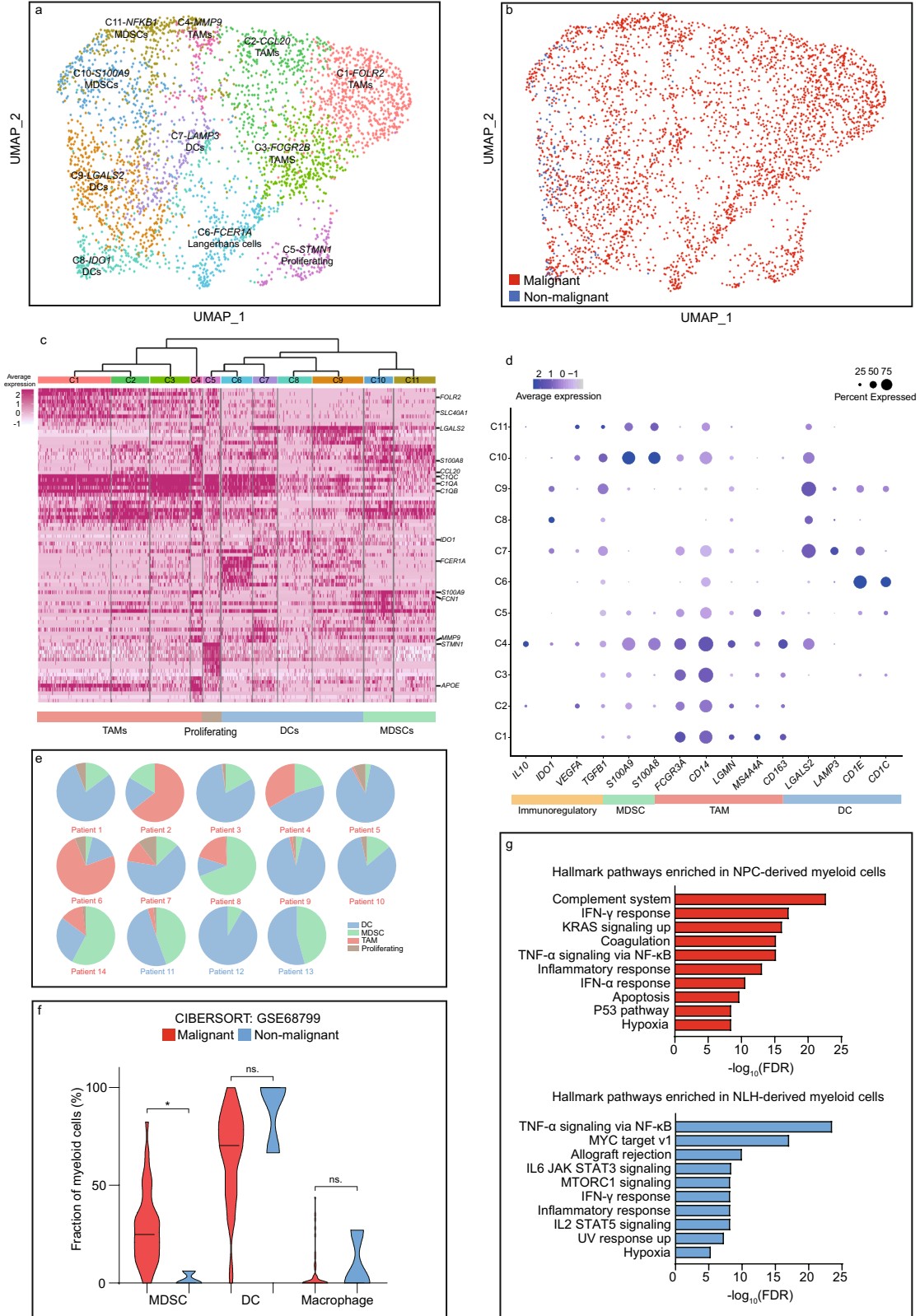

**Fig. 7 The NPC microenvironment harbors suppressive myeloid subtypes with high patient-specific heterogeneity. a** UMAP plot of 3,671 myeloid cells, color-coded by the associated cluster. **b** UMAP plot of 3,671 myeloid cells, color-coded by the sample type. **c** The expression matrix containing the top 50 enriched genes in each subpopulation defined in **a**, with a clustering tree categorizing related subpopulations into major subtypes. **d** The expression of functional signatures that were used to identified and characterized the subpopulations. **e** The inter-patient distribution of major myeloid subpopulations, shown by the percentage of total myeloid cells. **f** The estimated abundance of major myeloid subpopulations via CIBERSORTx (malignant $n = 42$ biologically independent samples, non-malignant $n = 3$ biologically independent samples). The two-sided Student's $t$ test was used to determine the statistical significance, $p$ values > 0.05 were represented as ns. (not significant), $p$ values ≤ 0.05 were represented as *. **g** The GSEA hallmark pathways enriched in NPC-derived and NLH-derived myeloid cells, ordered by −$\log_{10}$ FDR. Source data are provided as a Source Data file.

treatment outcome. The finding provides emerging opportunities for using these signatures as biomarkers and targets for early diagnosis and prognosis. Validating our findings in larger cohorts and clinical trials remains necessary to further elucidate the correlation between stromal composition and tumor progression. However, some of the stromal composition remains highly variable across the patients, and we can only confidently characterize the subtypes that are representative and enriched in the TME. The minor subpopulations residing in the NPC microenvironment might also have effects on altering the clinical outcome dramatically. Therefore, the in-depth identification and characterization of minor subpopulations in the TME remains necessary so that the single-cell sequencing technique should proceed to become capable of analyzing finer parts of the tumor mass.

Most notably, we confirmed that the presence of a dynamic transitional process from activated T cells to exhausted T cells in the TME. The exhausted T cells, instead of being completely dysfunctional, exhibited features that were associated with TLS formation via the B cell recruitment. Besides, we focused on the increased abundance of suppressive Tregs in the NPC microenvironment and found promising signatures associated with the Treg activation. The gene regulatory network reflected the importance of calcium channels in shaping the TME and affecting the immune response in NPC patients. Previous studies have reported that calcium channel served as a switch that could turn immune cells into the activated mode[48,49]. We also confirmed the prevalence of B cell subpopulations in the NPC microenvironment that has not been previously recognized. The presence of B-cell signatures and the abundance of B cell subpopulations were significantly correlated to the disease progression of NPC patients. We hypothesized that B cells might also play a vital role in regulating the outcome of radio-chemotherapy and immunotherapy. Moreover, the myeloid cells were mainly derived from the TME, indicating the recruitment of myeloid cells was an NPC-specific event. The MDSCs and TAMs in the TME served as a connecting bridge to facilitate immune suppression, since many signature genes in these subtypes have been previously found associated with exhaustion, cell cycle arrest of immune cells. In summary, the single-cell data from our clinical cohort and the bulk RNA data from the GEO cohort provide a vital and unique insight into the identification and characterization of stromal landscape in the NPC microenvironment and are likely to stimulate promising research in the future.

## Method

**Enrollment of NPC and NLH patients**. The study was approved by the ethics committee at the University of Hong Kong and the University of Hong Kong-Shenzhen Hospital. We complied with all related ethical regulations. Written informed consent was obtained from all patients with primary NPC and non-malignant NLH for their tissues to be used in this study.

**Preparation of the single-cell suspension**. After the endoscopic biopsy, the fresh tissue samples (2–3 mm³) from NPC and NLH patients were rinsed with phosphate-buffered saline (PBS) on ice. Subsequently, each sample was placed into the 500 µL dissociation medium containing 0.5 mg/mL collagenase IV (Sigma) and 1 mg/mL DNAse I (Sigma) in RPMI-1640 (ThermoFisher Scientific). Samples were minced in the dissociation medium on ice and then incubated for 30 min at 37 °C, with manual vortexing every 10 min. Then, 1 mL cold RPMI-1640 containing 10% fetal bovine serum (FBS, ThermoFisher Scientific) was added and each sample was filtered using 70-µm nylon mesh (ThermoFisher Scientific), and subsequently filtered again using 40-µm nylon mesh (ThermoFisher Scientific). The samples were centrifuged at 300×g for 5 min at 4 °C, and the supernatant was discarded. The single cells were resuspended in 1 mL ACK lysis buffer (ThermoFisher Scientific) and incubated for 5 min. 5 mL cold RPMI-1640 containing 10% FBS was added and the cell mixture was centrifuged at 300×g for 5 min at 4 °C. The single-cell pellet was resuspended in PBS without calcium and magnesium ions to reach the density ≤1000 cells/µL.

**Single-cell RNA and V(D)J sequencing**. The single-cell encapsulation and library preparation was done at the Centre for PanorOmic Sciences of the University of Hong Kong. The single-cell suspension was converted to uniquely barcoded RNA, TCR and BCR libraries by using the Chromium Single Cell 5′ Library, Gel Bead & Multiplex Kit, Chromium Single Cell V(D)J Enrichment Kit (Human T Cell), Chromium Single Cell V(D)J Enrichment Kit (Human B Cell), and Chromium Single Cell Chip Kit (10× Genomics), as per manufacturer's instructions. The libraries were sequenced on a NovaSeq 6000, and mapped to the human genome (hg38) using CellRanger (10× Genomics). Gene positions were annotated as per Ensembl build 85 and filtered for biotype (only protein-coding, long intergenic noncoding RNA, antisense, immunoglobulin, T-cell receptor, and B-cell receptor).

**Sample aggregation and determination of the batch effect**. The processed data of 14 samples were aggregated by cellranger aggregation function with default parameters, where the sequencing reads were normalized across samples. Since all libraries were generated by the same chemistry version and the data for all the samples were generated and processed at a similar time (within six months), no significant batch effect was observed in the aggregated data (Supplementary Figs. S8–S10).

**Quality control of the single-cell data and determination of major cell lineages**. The raw gene expression matrix generated from each sample were aggregated using CellRanger (version 3.1.0) and converted to a Seurat object using Seurat R package (version 3.0). The cells that had unique feature counts over 4000 or <200, or >15% mitochondrial counts, were filtered. From the remaining 66,627 cells, the gene expression dataset was normalized and subsequently dimensionally reduced based on 3,000 differentially expressed genes and principal components ($n = 25$). Major cell clusters projected in the two-dimensional UMAP representation were annotated to known cell lineages using well-recognized marker genes.

**Subclustering of the T cell, B cell, and myeloid populations**. To identify subpopulations within these major cell clusters, we performed the second-round UMAP reduction with the resolution of 1 on cells belonging to each of the major cell clusters separately. The number of principal components in each major subtype was independently determined by the Elbowplot function implemented in Seurat v3. In addition, the doublets in each cell cluster were identified by graph-based clustering and R package DoubletFinder[50] and filtered out from the analysis. The second-round UMAP reduction revealed 36 distinct subpopulations in total in the T cells, B cells, and myeloid cells.

**Clustering validation of T cell and B cell subpopulations via the random forest**. To validate the presence of the T cell and B cell subpopulations in the microenvironment, we collected additional tissues from patients 9-13 and analyzed them via the aforementioned single-cell sequencing protocol. We established a 37,442-cell cohort and clustered the dataset based on the same method used in the discovery cohort. To assess to which of the T and B cell subpopulations these cells correspond, we generated a Random Forrest classifier using modified BuildRF-Classifier for Seurat (version 3.0) in the discovery cohort and classify the cells based on this classifier by ClassifyCells in the validation cohort. The correlations between clusters in the discovery and validation cohort were determined. The random forest classifier assigned the 37,442 cells from the validation set of five patients to the 25T and B cell subpopulations identified in the discovery cohort of 14 patients.

**Identification of differentially expressed genes between the NPC and NLH microenvironment**. To identify the differentially expressed genes, we applied MAST analysis to assess genetic alterations between NPC-derived cells and NLH-derived cells, which can effectively eliminate the large deviation between cells in heterogenous single-cell sequencing data. The genes with false-discovery rate (FDR) ≤ 0.05 were considered significantly different in the NPC and NLH microenvironment. Significantly up-regulated genes were required to have an average expression in the NPC microenvironment that was at least ≥0.5 log₂-fold higher than the average expression in the NLH microenvironment.

**Survival correlation to the bulk RNA sequencing data in GEO**. To further investigate the survival outcomes of identified signature-genes in our single-cell sequencing data, we assessed their expression in bulk RNA sequencing data from GEO (GSE102349). There are 113 NPC patients enrolled in the study. Subsequently, the read counts were generated by HTSeq (version 0.9.1) and normalized by DESeq2 (version 1.22.2). The quality of the RNAseq data was evaluated by Picard metrics (version 2.17.4) and RSeQC (version 2.6.4). Only the samples with at least 60% of reads mapped to coding regions were included. One sample which failed to pass the quality control and 24 samples without complete follow-up data were removed from the downstream analysis. The remaining 88 patients were included for the subsequent survival analysis. The significance of progression-free survival based on signature genes and functional scores (binary: high vs. low) was evaluated by the two-sided log-rank test. To assess the correlations between functional scores and immune abundance with the prognosis of the NPC patients,

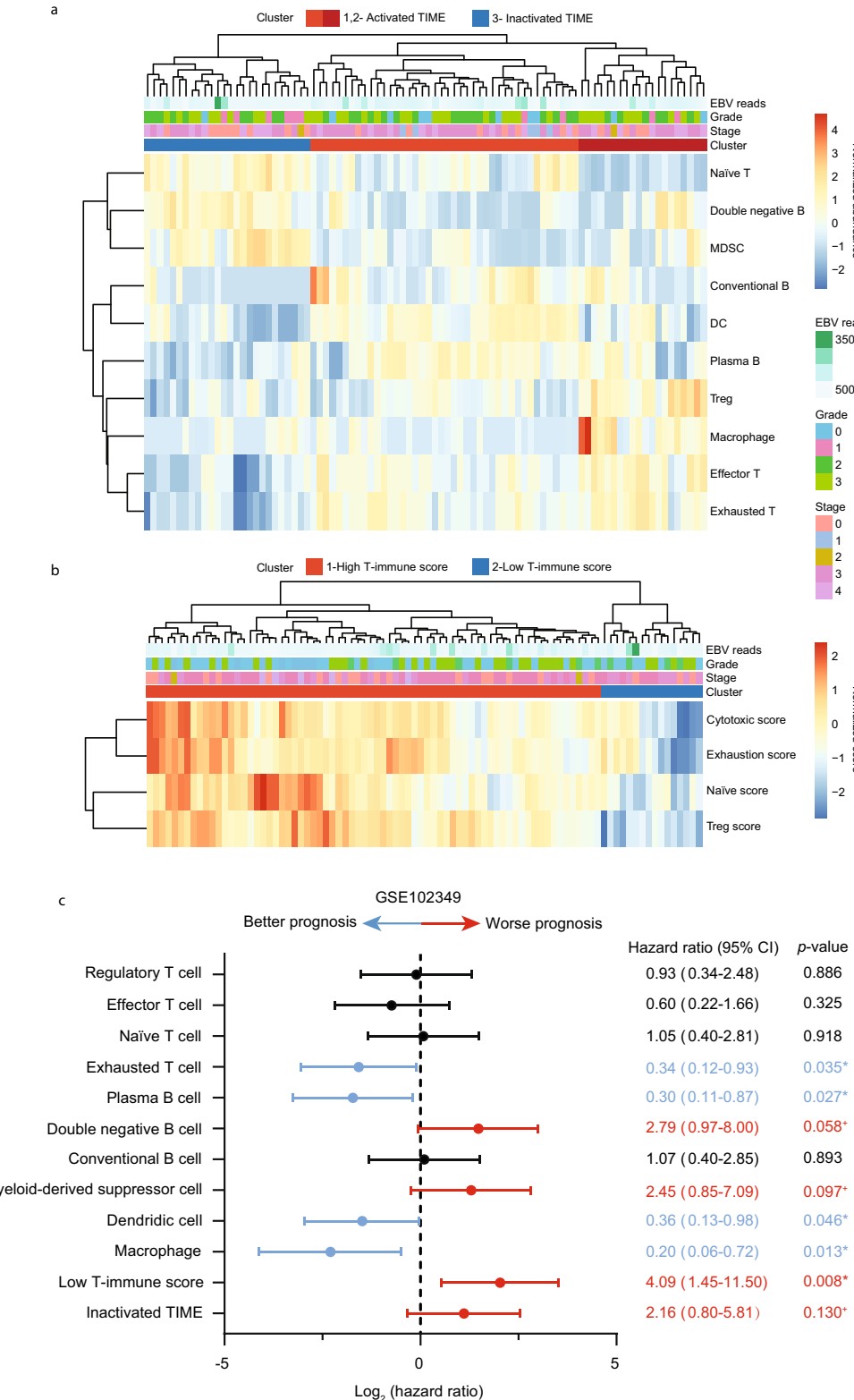

**Fig. 8 Deconvolution analysis and functional modules reveal dynamic immune signatures that are associated with patient prognosis. a** Heatmap of the normalized immune abundance and clinical parameters in 88 NPC patients estimated by CIBERSORTx. Patients were clustered into three groups, representing those with activated TIME (cluster 1 and 2) or inactivated TIME (cluster 3). **b** Heatmap of the normalized functional scores and clinical parameters in 88 NPC patients calculated by our linear model. Patients were clustered into two groups, representing those with high T-immune score (cluster) or low T-immune score (cluster 2). **c** The correlation between estimated subpopulations, patient clusters and progression-free survival in 88 NPC patients. The log₂ hazard ratio and the associated *p*-value was evaluated by the Cox proportional hazards model with 95% CI. *p* values ≤ 0.05 were considered statistically significant in prognosis, whereas *p* value > 0.05 and ≤ 0.15 were considered marginally significant in prognosis and represented as ⁺. Source data are provided as a Source Data file.

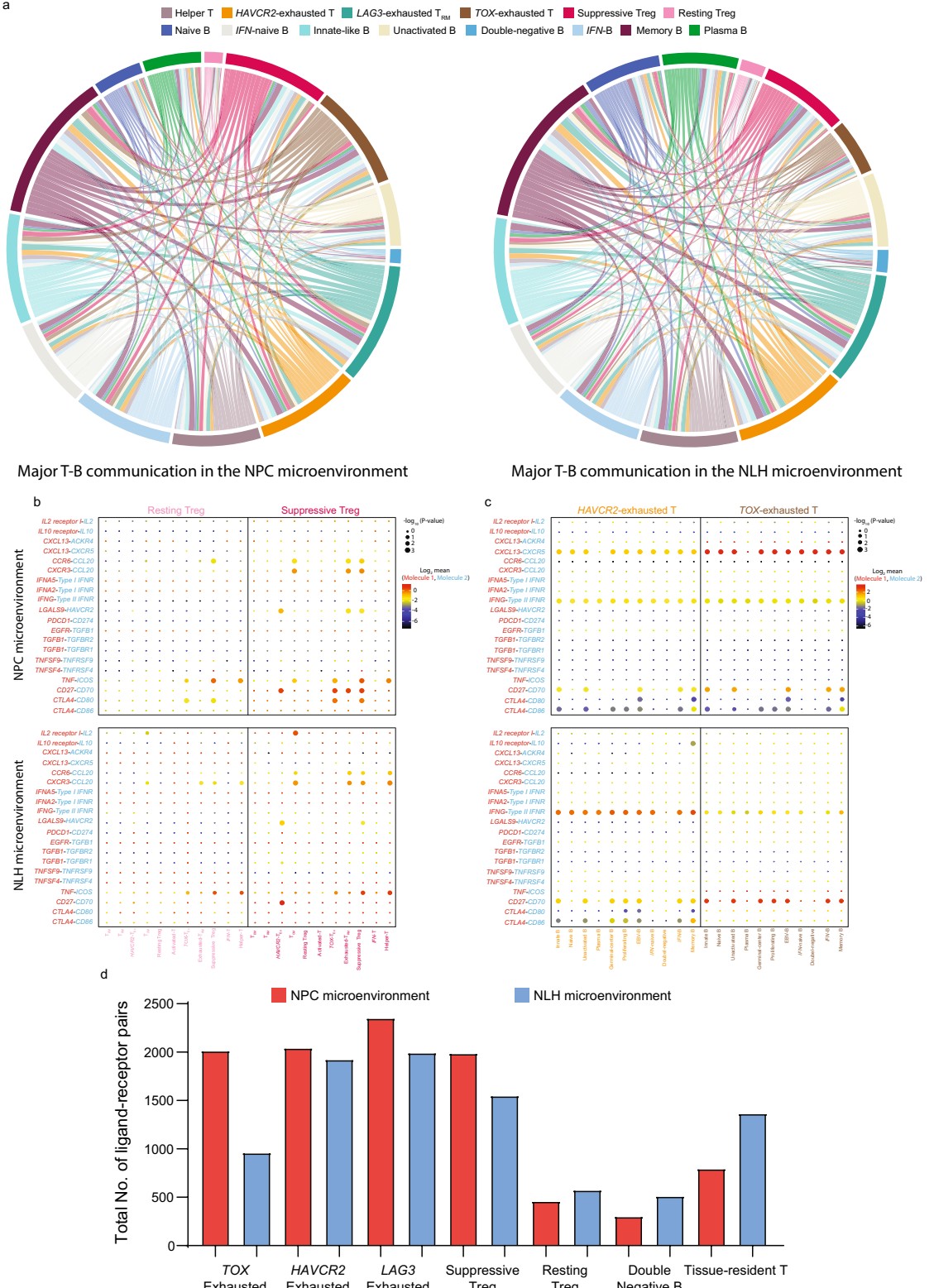

**Fig. 9 Single-cell referenced communication analysis exhibits tumor-specific ligand–receptor interactions within T and B cell subpopulations. a** The cell-cell communications within major T and B cell subpopulations in the NPC microenvironment and NLH microenvironment, shown by chord diagram, colored by cell subtypes. The size of the arc represented the total number of ligand–receptor pairs, and the size of the flow represented the number of ligand-receptor pairs between two cell subtypes. **b** Overview of selected ligand-receptor interactions between resting Treg/suppressive Treg and functional T cell subpopulations, **c** between *HAVCR2* exhausted T cells/*TOX* exhausted T cells and all B cell subpopulations. *p* values were calculated using the two-sided Student's *t* test with 95% CI and refer to the enrichment of the interacting ligand-receptor pair in each of the interacting pairs of cell types. Log$_2$ mean referred to the strength of a specific ligand-receptor connection in the corresponding cell types. **d** The total number of ligand-receptor pairs in the NPC-derived and NLH-derived T and B subpopulations. Source data are provided as a Source Data file.

we applied a Cox proportional hazards model in IBM SPSS Statistics 22 to evaluate the hazard ratio between these signatures and patient prognosis.

**Characterization of functional modules in T cells**. We identified the most representative genes associated with naïve (*CCR7*), cytotoxic (*NKG7*), exhausted (*LAG3*), and Treg (*FOXP3*) modules. We later constructed gene profiles containing the top ten genes that were mostly correlated to these representative signatures in naïve, cytotoxic, exhausted and Treg subpopulations. Based on these gene profiles, we subsequently established generalized linear models in naïve T cells, cytotoxic T cells, exhausted T cells, and Tregs using glm function in R version 3.6. During the linear model construction, we used the binomial family function because the response variable is categorical, indicating that the cells belong to certain clusters or not (1 stands for yes and 0 stands for no). The generalized linear models assigned a distinct parameter to each gene in the profile based on its expression weight. Subsequently, the functional scores were calculated based on the parameters estimated from the model, which represented the relative abundance of certain cell types.

**Construction of gene regulatory network**. Top 30 upregulated genes in the NPC-derived T cells and myeloid cells were extracted by MAST based on the aforementioned protocol. Their up-stream transcription factors were predicted in RegNetwork: Regulatory Network Repository (http://regnetworkweb.org/). Only the transcription factors with high confidence were collected. The genetic and transcriptional interaction was predicted by GeneMANIA, implemented in Cytoscape. The gene regulatory network was constructed using Cytoscape.

**Clonotype diversity and motif analysis**. We characterized the clonotypes of the TCR β chain and the BCR IGH chain in each patient. We used Shannon Index and dominance index (count of the maximum clone/sum count of all clones) to evaluate the diversity of clonotype in each patient. To find the motifs for CDR3 amino acid sequence, we applied MoDec tool to perform motif deconvolution in the T cells ad B cells derived from the NPC and NLH microenvironment, respectively. The optimal number of motifs (6) was selected based on the Akaike Information Criterion (AIC).

**Pseudotime developmental trajectory of T cell, B cell, and myeloid populations**. We applied the Monocle v2 to determine the potential development lineages between the T cell, B cell, and myeloid subpopulations. The differentially expressed genes across the clusters were identified by FindMarkers in Seurat for T cell, B cell, and myeloid subpopulations, respectively. The cells were ordered in pseudotime, where the best trajectory tree was fit after reduction of dimensionality of the data by Reversed Graph Embedding algorithm. The proliferating cluster was excluded from the analysis in three subpopulations because it is a heterogenous cluster containing varied subtypes of proliferating cells.

**Estimation of immune abundances from RNA-sequencing data via CIBERSORTx**. Based on our single-cell sequencing data, we selected several representative clusters in T cells (Naïve T, effector T, exhausted T and Treg), B cells (Plasma B, double-negative B and pan-B), and myeloid cells (MDSCs, DCs, and macrophages), respectively. We subsequently generate a gene expression matrix containing 733 genes that characterized these clusters. Using the signature matrix as the reference, CIBERSORTx (https://cibersortx.stanford.edu/) deconvoluted bulk RNA-sequencing data obtained from two GEO cohorts into the subpopulation abundances in each patient. Since the signature patterns of effector T cells and exhausted T cells were highly similar, CIBERSORTx could not adequately distinguish these two subtypes. Thus, we applied an unsupervised linear regression to calculate the relative percentage of effector and exhausted subpopulations based on CIBERSORTx data and functional scores. The deconvoluted information of each patient was used to validate our findings from single-cell sequencing and further correlated with progression-free survival in NPC patients.

**Constructing the molecular interaction network via CellPhoneDB**. CellPhoneDB (version 2.0.0, https://github.com/Teichlab/cellphonedb) used the cluster annotation and counts from our single-cell transcriptomics data to compute cell-cell communication within the identified cell subtypes. The default ligand-receptor pair information was used in this process. The $p$ value ≤ 0.05 indicated significant enrichment of the interacting ligand-receptor pair in each of the interacting pairs of cell subpopulations. Log$_2$ mean referred to the log$_2$-transformed total mean of the individual partner average expression values in the corresponding interacting pairs of cell subpopulations.

**Statistical analysis**. Statistical analysis was performed and plots were generated using GraphPad Prism 8 and IBM SPSS Statistics 22, and mentioned in the figure legends. All data points are shown for bar plots with a sample size <10. For larger sample sizes, violin plots are used to visualize the data distribution. Dara are presented as the mean values ± SD. For normally distributed data, $p$ values were evaluated by the two-sided Student's $t$ test. $p$ values > 0.05 were considered not statistically significant and represented as ns., $p$-values ≤0.05 were represented as *,

≤0.01 as **, ≤0.001 as ***, ≤0.0001 as ****. $p$ values were adjusted (FDR) for multiple hypothesis testing in the MAST analysis (implemented in R version 3.6) and GSEA analysis (GSEA v4.0.3). In survival analysis, $p$ values were evaluated by the two-sided log-rank test (survival analysis based on signature genes and functional scores) and the Cox proportional hazards model (hazard ratio).

## Data availability

The raw and processed single-cell sequencing data are publicly available in Gene Expression Omnibus (GEO) with the accession number GSE150825.

(https://www.ncbi.nlm.nih.gov/geo/query/acc.cgi?&acc=GSE150825)

The publicly available NPC bulk RNA sequencing data used in the study are available in GEO with the accession number GSE68799 and GSE102349.

(https://www.ncbi.nlm.nih.gov/geo/query/acc.cgi?&acc=GSE68799, https://www.ncbi.nlm.nih.gov/geo/query/acc.cgi?&acc=GSE102349)

The publicly available NPC single-cell RNA sequencing data used in the study are available in GEO with the accession number GSE150430.

(https://www.ncbi.nlm.nih.gov/geo/query/acc.cgi?acc=GSE150430)

The patient information for single-cell sequencing and bulk RNA sequencing is available in Supplementary Tables S1–S3. All data supporting the findings of the study are available within the Article, Supplementary Information or available from the corresponding author upon request. Source data are provided with this paper.

## Code availability

The codes used to perform clustering and functional analysis via Seurat is available on Github at https://github.com/LanqiGong/NPC-and-NLH-single-cell-analysis.

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

## Acknowledgements

This work was supported by grants from the Hong Kong Research Grant Council (RGC) grants including GRF (17143716), Collaborative Research Funds (C7065-18GF and C7026-18GF), Theme-based Research Scheme (T12-704/16-R), National Key Sci-Tech Special Project of Infectious Diseases (2013ZX10002-011-005), and The Shenzhen Peacock team project (KQTD2015033117210153 and KQTD2018041118502879), X.Y.G. is the Sophie YM Chan Professor in Cancer Research.

## Author contributions

X.G. designed and supervised the study. L.G., W.D., and X.G. wrote the manuscript. D.L.K., P.W., and S.L. supervised the sample collection. L.G., Q.Y., Y.Z., X.F., L.L., M.L., and B.L. performed the related experiments. L.G., W.D., B.Z., L.K.C., and Q.C. contributed to the data analysis. A.W.L. and Y.W. collected the clinical information and performed the pathological examination. J.H., V.H.L., K.L., Z.C. and A.W. Lee contributed to the interpretation of clinical information. All of the authors have read and approved the manuscript.

## Competing interests

The authors declare no competing interests.
