## [Peer Review File · Nature Communications]

Editorial Note: In the second round of review, Reviewers #1 and #2 were unavailable to comment on the revised manuscript. Therefore, Reviewer #4 was recruited to comment on the authors' response to these reviewers.

REVIEWER COMMENTS

Reviewer #1 (Remarks to the Author):

The authors present data and analysis from 104,069 cells from 14 patients of infiltrating stromal and immune cells in NPC. The data as collected is superb. Most of the conclusions are reasonable.

However, the paper suffers slightly by not explaining well how the authors determine a given cell is "exhausted". While I understand there is a gene expression signature that defines a cell as such, the WRITING of the paper jumps to the conclusion that it IS exhausted with NO reference to how this was concluded. Yes, I can see in the figures the genes that most likely call out the gene signature for a given cell subset, but there is a disconnect in how the paper is written as compared to the (very nice) details in the Figures.

Perhaps I am just picky, but the paper would benefit from carefully guiding the reader through some of the conclusions by exact referencing to the actual figures. The cell types in the UMAP would make a nice example. I should not have to go back and forth trying to determine how they came to certain conclusions. The Results is written at too high a level and needs to take a little more time/effort taking the reader through the figures.

In addition the manuscript in several places talked about correlation to outcomes, but never really explicitly called it out. For instance:

Line 123 TME-enriched subpopulations that exhibited dysfunctional features and exerted immunosuppressive functions that profoundly affected the prognosis of NPC patients.

This comes very early in the results section, but there is no supporting Figure that has been presented yet that leads me to want to believe what the authors just stated. This is perhaps the most interesting part of the manuscript, but it seems to be hidden in one figure, though referenced in several places, and as I noted, referenced early before even showing the results.

This is one example (of several) wherein the paper seems disjointed in how it is laid out. The results are all there, I actually BELIEVE the results. It's just that the presentation, timing of where and when things are talked about, seems a little haphazard.

There are numerous intriguing observations in the paper, and the data will eventually serve as an important contribution to the field.

My recommendation is that the paper eventually is accepted, but that significant reworking of the manuscript (no new experiments) be done to lay out the story more succinctly.

Reviewer #2 (Remarks to the Author):

Comprehensive single-cell sequencing reveals the composition and dynamics of stromal cells in tumor microenvironment of NPC by Xin-Yuan Guan.

This study by Guan and colleagues provides a new next generation sequencing approach to generating information about nasopharyngeal carcinomas and this information is additional data that explores the changes in the heterogenic and dynamic nature of cancers including NPCs. Previous sequencing technologies have provided some ideas about the landscape of the stromal cells in NPCs but there are still quite a long way to go in understanding the makeup of the NPC microenvironment that may determine the severity of the disease as well as the immune response. The authors employed single-cell RNA-Seq of over 66 thousand cells from 14 individual patients and integrated these with additional identification of T and B cells in the tumor microenvironment. The compared with sequencing results from nonmalignant microenvironment. The results were interesting as their analyses identified 5 stromal clusters and 36 subpopulations based on genetic profiles. The sequencing data suggests that there are representative features in the TME of NPCs which may affect progression and therapeutics. The authors then used bulk data from 88 RNA-Seq data to correlate some characteristics and survival which were progression free of disease. Although the data is very interesting and provides new information I could not find any clear mechanistic studies that were built of the studies and further validation of the scRNA-Seq data. This paper is simple a manuscript about the analyses that provides data that provides information that may be useful for our understanding of NPC characteristics. However, whether the findings based on scRNA-Seq is truly linked to any phenotype or disease based on mechanism that explains how it works is not provided and therefore the manuscript falls short of helping the field besides adding new large data sets. It should also be mentioned that there have been many published papers on NPC sequencing and how these large data sets are used and validated even from individuals in the same group has not been done. Has these data been compared to the many other data sets that are already present in public databases.

Here are my comments for consideration

1. Large data sets that are useful and likely provide new information.
2. The data obtained have not been validated in any way with typical proven methodologies that will be convincing to other in the field.
3. Now correlations with other large data sets available on public databases.
4. Unsure if any of this information is reproducible as it was not clear.
5. Some statements in the abstract is different from what is present in the body of the text.
6. The functional modules that were identified from sequencing based on T and Tregs are interesting but it would be important to validate some of these with actual validated data to confirm their findings especially for the NKG7, CCR7, LAG3 and FOXP3 positive cell population.
7. It may be important for the authors to focus on some of the key findings are carefully validate them.

Reviewer #3 (Remarks to the Author):

In this interesting manuscript " Comprehensive single-cell sequencing reveals the composition and dynamics of stromal cells in the tumor microenvironment of nasopharyngeal carcinoma", Dr. Gong and colleagues investigated the tumor infiltrating stromal cells landscape of NPC. The data generated is interesting and could be a good addition to the field. However, previous similar publications have decreased its novelty.

Major comments:

1. A recent paper on single cell profiling of tumor cells and infiltrating immune cells of NPC published in 2020 by Jin Zhao et al. in Cancer Letters. I don't think the authors cited and discussed the similar and novel findings compared to that study.
2. How about the inter-patient heterogeneity? Did the TME components, NPC cells, interaction between subsets of immune, between immune cells and NPC cells similar or different in each patient? Is there any trend that age, gender, stage etc. impacts these relationships?
3. It will be much convincing if some of the subpopulations can be confirmed by multiplex IF or MIBI or IMC.

Minor comments:

1. Patients clinical information should be provided for both the 14 and the 83 patients.
2. For the analysis of the 83 patients, was the survival adjusted for other prognostic parameters?
3. Was EBV-positivity confirmed by PCR or antibody?
4. How were the malignant cells defined from the biopsy?
5. How did you define malignant vs non-malignant cells? Pathology? Gene expression? Why do you think they merge together?
6. The fraction of malignant cell in Fig. 1b appeared to be quite different (proportion wise) from that in Fig.1f. Can you explain why?
7. How did you adjust batch effect? Please clarify that in the method and a supp Fig showing whether batch effect was present and corrected may be helpful.
8. Line 170, what is the cut-off to define high vs low CXCL13?
9. Figure 6a, check the name of cluster 5.
10. Line 320, the results did not seem to clearly show the 'interpatient variation' of myeloid cells.
11. Line 322, no figure 6f, please double check it.
12. Line 339, describe briefly the potential therapeutic target by scRNA-seq analysis of NK cell and fibroblast. What approaches did authors suggest?
13. Line 346, what is the relationship of collagens type I-VI and LUM and FN1?

Dear Reviewer #1,

Reviewer #1 (Remarks to the Author):

The authors present data and analysis from 104,069 cells from 14 patients of infiltrating stromal and immune cells in NPC. The data as collected is superb. Most of the conclusions are reasonable.

- 1. However, the paper suffers slightly by not explaining well how the authors determine a given cell is “exhausted”. While I understand there is a gene expression signature that defines a cell as such, the WRITING of the paper jumps to the conclusion that it IS exhausted with NO reference to how this was concluded. Yes, I can see in the figures the genes that most likely call out the gene signature for a given cell subset, but there is a disconnect in how the paper is written as compared to the (very nice) details in the Figures.*

Our reply:

Thank you very much for your positive feedback on our single-cell data. Based on your suggestions, we have revised and reorganized the paragraphs that discussed T cell exhaustion. The revised paragraphs in the section “Identification and characterization of T-cell diversity in the NPC and non-malignant microenvironments”, contained more details and references that we used to characterize the exhausted subpopulation. For example, we have more specifically described how we identified exhausted T cell subpopulations based on marker genes and functional signatures. (“three distinct exhausted T cell subpopulations were identified and characterized by LAG3, HAVCR2 and TOX, which were previously reported in T cell dysfunction¹⁹⁻²¹.”)

Perhaps I am just picky, but the paper would benefit from carefully guiding the reader through some of the conclusions by exact referencing to the actual figures. The cell types in the UMAP would make a nice example. I should not have to go back and forth trying to determine how they came to certain conclusions. The Results is written at too high a level and needs to take a little more time/effort taking the reader through the figures.

In addition the manuscript in several places talked about correlation to outcomes, but never really explicitly called it out. For instance:

- 2. Line 123 TME-enriched subpopulations that exhibited dysfunctional features and exerted immunosuppressive functions that profoundly affected the prognosis of NPC patients.
This comes very early in the results section, but there is no supporting Figure that has been presented yet that leads me to want to believe what the authors just stated. This is perhaps the most interesting part of the manuscript, but it seems to be hidden in one figure, though referenced in several places, and as I noted, referenced early before even showing the results.*

Our reply:

We changed the sentence “TME-enriched subpopulations that exhibited dysfunctional features and exerted immunosuppressive functions that profoundly affected the prognosis of NPC patients” into “To further define the phenotype, functional state and prognostic value of the major clusters, we performed more in-depth characterization and validation within these genetically distinct subpopulations.” In addition, we have added a new section called “Correlation between stromal abundance and patient survival and molecular interaction in the infiltrating stromal subtypes” in the result part, which more specifically discussed with the correlation between immune infiltration and prognosis of NPC patients.

3. *This is one example (of several) wherein the paper seems disjointed in how it is laid out. The results are all there, I actually BELIEVE the results. It's just that the presentation, timing of where and when things are talked about, seems a little haphazard.*

Our reply:

Thank you very much for your recognition of our findings. We have added more details (in figures and in paragraphs) to identify and characterize cell subpopulations identified in our single-cell sequencing data. We also modified the flow of our manuscript in order to make it more readable for a broad audience with different backgrounds.

There are numerous intriguing observations in the paper, and the data will eventually serve as an important contribution to the field.

My recommendation is that the paper eventually is accepted, but that significant reworking of the manuscript (no new experiments) be done to lay out the story more succinctly.

We would like to first thank you very much for your positive feedback on our work, and we have significantly modified the flow of our manuscript to make the contents more readable and conclusions clearer.

Dear Reviewer #2,

Reviewer #2 (Remarks to the Author):

Comprehensive single-cell sequencing reveals the composition and dynamics of stromal cells in tumor microenvironment of NPC by Xin-Yuan Guan.

This study by Guan and colleagues provides a new next generation sequencing approach to generating information about nasopharyngeal carcinomas and this information is additional data that explores the changes in the heterogenic and dynamic nature of cancers including NPCs. Previous sequencing technologies have provided some ideas about the landscape of the stromal cells in NPCs but there are still quite a long way to go

in understanding the makeup of the NPC microenvironment that may determine the severity of the disease as well as the immune response.

The authors employed single-cell RNA-Seq of over 66 thousand cells from 14 individual patients and integrated these with additional identification of T and B cells in the tumor microenvironment. The compared with sequencing results from non-malignant microenvironment. The results were interesting as their analyses identified 5 stromal clusters and 36 subpopulations based on genetic profiles. The sequencing data suggests that there are representative features in the TME of NPCs which may affect progression and therapeutics. The authors then used bulk data from 88 RNA-Seq data to correlate some characteristics and survival which were progression free of disease.

Although the data is very interesting and provides new information I could not find any clear mechanistic studies that were built of the studies and further validation of the scRNA-Seq data. This paper is simple a manuscript about the analyses that provides data that provides information that may be useful for our understanding of NPC characteristics. However, whether the findings based on scRNA-Seq is truly linked to any phenotype or disease based on mechanism that explains how it works is not provided and therefore the manuscript falls short of helping the field besides adding new large data sets. It should also be mentioned that there have been many published papers on NPC sequencing and how these large data sets are used and validated even from individuals in the same group has not been done. Has these data been compared to the many other data sets that are already present in public databases.

Here are my comments for consideration

- 1. Large data sets that are useful and likely provide new information.*

Our reply:

Thank you very much for your recognition of our single-cell data.

- 2. The data obtained have not been validated in any way with typical proven methodologies that will be convincing to other in the field.*

Our reply:

According to your suggestions, we have further validated our findings, including the relative abundance of TME-enriched subpopulations and functional modules, in two independent GEO cohorts (GSE68799 and GSE102349, containing the RNA-seq data from 45 patients and 83 patients, respectively; data shown in fig. 2c, supplementary fig. 3d, fig. 5e and fig. 6f). We also validated the enrichment of exhausted T cells and Tregs via multicolor immunofluorescence of CD3, CD8, FOXP3 and PD-1, showing a consistent result with our single-cell data (Fig. 2e).

- 3. Now correlations with other large data sets available on public databases.*

Our reply:

In the revised manuscript, we have used devolution analysis to estimate the abundance of major immune subpopulations in 83 NPC patients from GSE102349, and correlated the immune abundance with progression-free survival (Fig. 7a and 7b). The functional modules which were validated in GSE68799 and GSE102349, were also linked with patient survival in GSE102349 (Supplementary fig. 3f).

4. *Unsure if any of this information is reproducible as it was not clear.*

Our reply:

Our findings, such as stromal compositions and immune signature analysis, are highly consistent with the results from recent single-cell studies on NPC, indicating the reliability and reproducibility of our single-cell data (Zhao et al. *Cancer Letters*, 2020 May; Chen et al. *Cell Research*, 2020 July). Meanwhile, our single-cell data provided additional information such as clonal analysis, deconvolution analysis and comparison between immune cells derived from TME and non-malignant microenvironment. In addition, we have validated the T cell and B cell clustering in an independent single-cell cohort (Supplementary fig. 1c-f).

5. *Some statements in the abstract is different from what is present in the body of the text.*

Our reply:

The abstract has been modified to clearly convey the key information in our manuscript;

6. *The functional modules that were identified from sequencing based on T and Tregs are interesting but it would be important to validate some of these with actual validated data to confirm their findings especially for the NKG7, CCR7, LAG3 and FOXP3 positive cell population.*

Our reply:

The functional modules were further validated in GSE68799 and GSE102349 using spearman's correlation (Supplementary fig. 3d). Both cohorts showed a very strong correlation between functional scores and the corresponding genes, indicating that our functional modules can comprehensively describe the dynamic functional state of NKG7, CCR7, LAG3 and FOXP3 positive cell populations.

7. *It may be important for the authors to focus on some of the key findings are carefully validate them.*

Our reply:

We will continue to investigate our key findings in experimental settings (in vitro function assays, animal models, etc.), especially to validate the function and molecular mechanism of dysfunctional and immunoregulatory subtypes in NPC microenvironment.

Dear Reviewer #3,

Reviewer #3 (Remarks to the Author):

In this interesting manuscript “Comprehensive single-cell sequencing reveals the composition and dynamics of stromal cells in the tumor microenvironment of nasopharyngeal carcinoma”, Dr. Gong and colleagues investigated the tumor infiltrating stromal cells landscape of NPC. The data generated is interesting and could be a good addition to the field. However, previous similar publications have decreased its novelty.

Major comments:

- 1. A recent paper on single cell profiling of tumor cells and infiltrating immune cells of NPC published in 2020 by Jin Zhao et al. in Cancer Letters. I don't think the authors cited and discussed the similar and novel findings compared to that study.*

Our reply:

We have cited the paper you mentioned in the discussion part (the first paragraph). Additionally, another paper published in Cell Research in July by Chen et al. was also cited and discussed in our manuscript. The immune composition and abundance identified and characterized by Jin Zhao et al., and Chen et al. were highly consistent with our results; The novelty of our findings include clonal analysis, deconvolution analysis and comparison between immune cells derived from TME and non-malignant microenvironment.

- 2. How about the inter-patient heterogeneity? Did the TME components, NPC cells, interaction between subsets of immune, between immune cells and NPC cells similar or different in each patient? Is there any trend that age, gender, stage etc. impacts these relationships?*

Our reply:

We have shown the inter-patient heterogeneity in T cell, B cell and myeloid cell subpopulations, respectively (Supplementary fig. 3e, fig. 5c and fig. 6e). There existed minor inter-patient heterogeneity in T cell, B cell and myeloid cell subpopulations. Only TOX-exhausted T cells, plasma B cells and TAMs exhibited high inter-patient heterogeneity and we discussed the phenomenon in our manuscript. We also used CellPhoneDB to construct a molecular interaction network within all identified cell subtypes in our single-cell data (Fig. 7c and supplementary fig 8); Because our single-cell cohort only contained 14 patients, we did not observe any clinical parameter that was correlated to stromal interaction in the NPC and NLH microenvironment.

- 3. It will be much convincing if some of the subpopulations can be confirmed by multiplex IF or MIBI or IMC.*

Our reply:

The exhausted T cell subpopulation (CD8⁺ PD1⁺) and Treg subpopulation (CD3⁺ FOXP3⁺) in the NPC and NLH microenvironment were validated using multicolor immunofluorescence (Fig. 3e). The result showed that the exhausted T cell subpopulation and Treg subpopulation were highly enriched in the TME.

Minor comments:

1. *Patients clinical information should be provided for both the 14 and the 83 patients.*

Our reply:

The clinical information for patients participated in the present study and two independent GEO cohorts was provided as supplementary tables (Supplementary table 1-3).

2. *For the analysis of the 83 patients, was the survival adjusted for other prognostic parameters?*

Our reply:

No other prognostic parameters were involved in adjusting the progression-free survival in 83 NPC patients.

3. *Was EBV-positivity confirmed by PCR or antibody?*

Our reply:

EBV-positivity was confirmed by the EBV-specific antibody.

4. *How were the malignant cells defined from the biopsy?*

Our reply:

The endoscopic biopsy was performed by professional surgeons who identified and located the NPC tumor (neoplasm) in the nasopharyngeal cavity and subsequently collected a small piece of tumor tissue for single-cell sequencing and pathological examination.

5. *How did you define malignant vs non-malignant cells? Pathology? Gene expression? Why do you think they merge together?*

Our reply:

The malignant cells were distinguished based on copy number variation. The malignant and non-malignant cells were merged during clustering because they expressed some mutual epithelial markers such as EpCAM and KRT19.

6. The fraction of malignant cell in Fig. 1b appeared to be quite different (proportion wise) from that in Fig.1f. Can you explain why?

Our reply:

The NPC tumor always had very high stromal infiltration, thus the immune cells obtained from the biopsy were constantly outnumbered the tumor cells. Secondly, during single-cell dissociation and sequencing, tumor cells were more likely to undergo cell apoptosis compared to immune cells.

7. How did you adjust batch effect? Please clarify that in the method and a supp Fig showing whether batch effect was present and corrected may be helpful.

Our reply:

The analysis result of 14 samples was aggregated by cellranger aggregation function with default parameters, where the sequencing reads were normalized across samples. Since all libraries were generated by the same chemistry version and the data for all the samples were generated and processed at a similar time (within six months), no significant batch effect was observed in the aggregated data. For example, fig. 1b showed that the cells derived from different patients were well-intermixed within all clusters.

8. Line 170, what is the cut-off to define high vs low CXCL13?

Our reply:

We used the median cut-off.

9. Figure 6a, check the name of cluster 5.

Our reply:

We have changed the name of cluster 5 into proliferating.

10. Line 320, the results did not seem to clearly show the ‘interpatient variation’ of myeloid cells.

Our reply:

We added a pie graph showing the inter-patient heterogeneity of major myeloid subpopulations; (Fig. 6e).

11. Line 322, no figure 6f, please double check it.

Our reply:

We have modified the figures based on your feedback.

12. Line 339, describe briefly the potential therapeutic target by scRNA-seq analysis of NK cell and fibroblast. What approaches did authors suggest?

Our reply:

We have changed the section title into “Fibroblasts and NK cells as microenvironment-based targets in NPC treatment.” We described that the therapeutics targeting NPC-specific ECM components might be useful to alter ECM-dependent NPC progression. Therapeutics that could stimulate the expansion of NK cells in the NPC microenvironment might be also feasible since we have found NK cells in the NPC microenvironment did not express exhaustion-associated markers.

13. Line 346, what is the relationship of collagens type I-VI and LUM and FN1?

Our reply:

We have modified this section, and we do not focus on the relationship between varied ECM components in the TME.

REVIEWER COMMENTS

Reviewer #3 (Remarks to the Author):

The authors have addressed my comments.
Thanks for the efforts.

Reviewer #4 (Remarks to the Author):

6. The functional modules that were identified from sequencing based on T and Tregs are interesting but it would be important to validate some of these with actual validated data to confirm their findings especially for the NKG7, CCR7, LAG3 and FOXP3 positive cell population.

Our reply: The functional modules were further validated in GSE68799 and GSE102349 using spearman's correlation (Supplementary fig. 3d). Both cohorts showed a very strong correlation between functional scores and the corresponding genes, indicating that our functional modules can comprehensively describe the dynamic functional state of NKG7, CCR7, LAG3 and FOXP3 positive cell populations.

Comment 6 response from Reviewer #3

These scatter plots provide some, but not very compelling validation. The genes whose expression is plotted on the vertical axes of these plots will be prominent components of the functional scores plotted on the horizontal axes, the other genes (presumably) being ones that were highly correlated with the one on the vertical axes. I would say that this conclusion is almost inevitable.

Some further comments.

1.. Several of the methods in the paper are not fully explained. For example, there is no explanation or detail about the scores. apart from this remark Based on these gene profiles, we subsequently established linear models to calculate the program-dependent scores in each cluster and patient. I think the reader deserves more: and explanation of the method, and details of the score. Otherwise, the results in this paper cannot be reproduced.

2.. I was extremely surprised to see the remark Since all libraries were generated by the same chemistry version and the data for all the samples were generated and processed at a similar time (within six months), no significant batch effect was observed in the aggregated data. This is at odds with all my experience. I note that nowhere in the ms are the cells for the 14 different patients labelled (e.g. within cluster) in a way that might support this statement.

3.. A small but concerning point is the following. In the plots displaying TCR (Supp Fig 4c) and BCR (Supp Fig 6b) motifs, all we see are "motifs" representing the first 4 and the last 2 or 3 amino acids. The start or end of TCR sequence are common motifs, e.g. CGA, CAT, CTC, CAS, YF, FF.... The starts of CDR3 beta chains have multiple possible motifs, but the end of the sequences normally have "YF" or "FF". The TCR clusters shown just coincidentally to have "CASS" as the start and "YF" or "FF" at the end. Unlike the tool (MoDec) that the authors used, GLIPH (<http://50.255.35.37:8080/>) does not use the first three and the last three amino acids as k-mer motifs for clustering, because they do not interact with antigens from the crystal structure

Dear Reviewer #4,

Comment 6 response from Reviewer #3

These scatter plots provide some, but not very compelling validation. The genes whose expression is plotted on the vertical axes of these plots will be prominent components of the functional scores plotted on the horizontal axes, the other genes (presumably) being ones that were highly correlated with the one on the vertical axes. I would say that this conclusion is almost inevitable.

Our reply:

Thank you very much for your feedback. In our discovery cohort (containing 14 samples) and validation cohort (containing 5 samples), we performed correlation analysis between functional scores and corresponding abundances of T cell subpopulations, including NKG7⁺ cytotoxic T cells, exhausted T cells, regulatory T cells and naïve T cells (Fig. 4a). The result showed that the functional scores and subpopulation abundances are highly correlated, suggesting our linear model can comprehensively depict the dynamic functional state in the NPC microenvironment. In order to further validate the universality of our linear model, we analyzed the T cell subpopulations of single-cell NPC data from Chen, Y.P. et al. (Chen, Y.P. *et al.* Single-cell transcriptomics reveals regulators underlying immune cell diversity and immune subtypes associated with prognosis in nasopharyngeal carcinoma. *Cell Res* (2020)). We calculated the functional scores for each sample using our linear model and found that the scores were highly correlated to cytotoxic, exhausted, naïve and regulatory abundances (Fig. 4b). The functional scores were significantly correlated to the corresponding genes, showing that the gene profiles we constructed based on single-cell data remained valid in NPC RNA-seq data (Fig. 4c). The RNA-seq analysis further indicated that the linear model is capable of more comprehensively and accurately quantifying real dynamic T cell functional states from bulk RNA sequencing data, compared to the signature-based characterization. The validation from our discovery cohort, validation cohort and NPC single-cell cohort from another study group provides a strong rationale that our linear model can accurately reflect the dynamic functional states of T cells in the NPC microenvironment and is useful to applied into the analysis of bulk RNA sequencing.

1.. Several of the methods in the paper are not fully explained. For example, there is no explanation or detail about the scores. Apart from this remark Based on these gene profiles, we subsequently established linear models to calculate the program-dependent scores in each cluster and patient. I think the reader deserves more: and explanation of the method, and details of the score. Otherwise, the results in this paper cannot be reproduced.

Our reply:

Thank you very much for your feedback. We have revised the method section in order to convey a precise and reproducible method to the audience. For example, we have written that “During the linear model construction, we used the binominal family function because the response variable is categorical, indicating that the cells belong to certain clusters or not (1 stands for “yes” and 0 stands for “no”). The generalized linear models assigned a distinct parameter to each gene in the profile based on its expression weight. Subsequently, the functional scores were calculated based on the parameters estimated from the model, which represented the relative abundance of certain cell types”. We also added a more detailed description of our linear model construction process in the main text under the section “Functional modules, clonal analysis and developmental trajectory revealed the dynamics of exhausted T cells and Tregs in the NPC microenvironment.”

2.. I was extremely surprised to see the remark Since all libraries were generated by the same chemistry version and the data for all the samples were generated and processed at a similar time (within six months), no significant batch effect was observed in the aggregated data. This is at odds with all my experience. I note that nowhere in the ms are the cells for the 14 different patients labelled (e.g. within cluster) in a way that might support this statement.

Our reply:

According to your suggestions, we have plotted the patient distribution on T cells, B cells and myeloid cells, respectively. We added these graphs as supplementary figures in our manuscript (Supplementary fig. 3d, supplementary fig. 5b and supplementary fig. 7a). We observed that most cells from the 14 patients were intermixed very well within clusters. There existed two patient-specific clusters, which were TOX-exhausted T cells (patient 9) and four subtypes of macrophages (patient 6 and patient 8). Although TOX-exhausted T cells were primarily derived from patient 9, other T cell subpopulations, B cells and myeloid cells from patient 9 were intermixed very well with the cells from other patients within clusters. Similarly, other cells from patient 6 and patient 8 were also intermixed very well within clusters. Therefore, we have not considered that there exists a significant batch effect in our single-cell data.

3.. A small but concerning point is the following. In the plots displaying TCR (Supp Fig 4c) and BCR (Supp Fig 6b) motifs, all we see are “motifs” representing the first 4 and the last 2 or 3 amino acids. The start or end of TCR sequence are common motifs, e.g. CGA, CAT, CTC, CAS, YF, FF.... The starts of CDR3 beta chains have multiple possible motifs, but the end of the sequences normally have “YF” or “FF”. The TCR clusters shown just coincidentally to have “CASS” as the start and “YF” or “FF” at the end. Unlike the tool (MoDec) that the authors used, GLIPH (<http://50.255.35.37:8080/>) does not use the first three and the last three amino acids as k-mer motifs for clustering, because they do not interact with antigens from the crystal structure

Our reply:

Thank you very much for your suggestion. As you suggested, we have excluded the first three and the last three amino acids of CDR3 sequences as k-mer motifs for re-clustering. The plots displaying TCR (Supplementary fig. 4c) and BCR (Supplementary fig. 6b) motifs logo were also updated.

b

NPC-enriched BCR motifs

NLH-enriched BCR motifs

REVIEWER COMMENTS

Reviewer #4 (Remarks to the Author):

Regarding point 1 of your rebuttal, the correct name for the family you used is "binomial", not "binominal".

Regarding point 2 on batch effects, a more illuminating and potentially insightful examination of this issue (with so many patient samples and cell types) would have been a carefully constructed series of UMAP plots organized by cell type and patient.

Dear Reviewer #4,

Regarding point 1 of your rebuttal, the correct name for the family you used is "binomial", not "binominal".

Our reply:

Thank you very much for pointing out our typo. We have changed it into the correct word in our manuscript.

Regarding point 2 on batch effects, a more illuminating and potentially insightful examination of this issue (with so many patient samples and cell types) would have been a carefully constructed series of UMAP plots organized by cell type and patient.

Our reply:

Thank you very much for your feedback. As you suggested, we have added UMAP plots to show the patient distribution in individual cell subtypes in T, B and myeloid cells. We have also added UMAP plots to show the T, B and myeloid cell distribution in each patient (Supplementary fig. 8-10).

REVIEWERS' COMMENTS

Reviewer #4 (Remarks to the Author):

Thank you for correcting the typo and providing the additional supplementary figures. They show negligible signs of batch=patient effects, as previously stated.

Dear Reviewer #4,

Regarding point 1 of your rebuttal, the correct name for the family you used is "binomial", not "binominal".

Our reply:

Thank you very much for pointing out our typo. We have changed it into the correct word in our manuscript.

Regarding point 2 on batch effects, a more illuminating and potentially insightful examination of this issue (with so many patient samples and cell types) would have been a carefully constructed series of UMAP plots organized by cell type and patient.

Our reply:

Thank you very much for your feedback. As you suggested, we have added UMAP plots to show the patient distribution in individual cell subtypes in T, B and myeloid cells. We have also added UMAP plots to show the T, B and myeloid cell distribution in each patient (Supplementary fig. 8-10).